# Burst of gyrification in the human brain after birth

Angeline Mihailov [1] ✉, Alexandre Pron[1], Julien Lefèvre[1], Christine Deruelle[1], Béatrice Desnous[2], Florence Bretelle[3], Aurélie Manchon[1,4], Mathieu Milh [2], François Rousseau[5], Nadine Girard [1,4,6] & Guillaume Auzias [1,6]

Gyrification, the intricate folding of the brain's cortex, begins mid-gestation and surges dramatically throughout the perinatal period. Yet, a critical factor has been largely overlooked in neurodevelopmental research: the profound impact of birth on brain structure. Leveraging the largest known perinatal MRI dataset—819 sessions spanning 21 to 45 postconceptional weeks—we reveal a burst in gyrification immediately following birth (~37 weeks post-conception), amounting to half the entire gyrification expansion occurring during the fetal period. Using state-of-the-art, homogenized imaging processing tools across varied acquisition protocols, and applying a regression discontinuity design approach that is novel to neuroimaging, we provide the first evidence of a sudden, birth-triggered shift in cortical development. Investigation of additional cortical features confirms that this effect is uniquely confined to gyrification. This finding sheds light onto the understanding of early brain development, suggesting that the neurobiological consequences of birth may hold significant behavioral and physiological relevance.

The perinatal period is generally defined as the time shortly after conception up until a few weeks to 1 year after birth[1,2]. Gathering participants to populate this period, in the form of pregnant mothers before birth and newborns after birth, is challenging due to the sensitive nature of these populations. Furthermore, characterizing perinatal brain development is a complex process since multifaceted and intricate genetic, molecular, and cellular mechanisms act in concert during brain maturation[3]. Nonetheless, this period is particularly critical since early alterations in developmental trajectories have been associated with later neurodevelopmental and physiological disorders[4–8].

Folding patterns in the healthy human brain begin forming around 20 weeks of gestation and expand drastically during fetal neurodevelopment, effectively morphing an initially smooth cortex into a convoluted structure[3,9]. Gyrification evolution during the fetal period has been characterized using Magnetic Resonance Imaging (MRI) in typically developing populations, with three waves of convolutions: primary and secondary that form during the third trimester, and tertiary, forming late in gestation and around the time of birth[10–16]. The gyrification index exhibits a drastic increase with gestational age starting in the second trimester, with an acceleration during the third trimester alongside the appearance of secondary and tertiary folds[13,17]. With respect to post-birth folding maturation, the first years of life typically show a *progressive slowdown* of gyrification as

global folding patterns become more established and stable[18,19]. Furthermore, gyrification during the first years of life correlates positively with brain volume and exhibits differences in sex[18].

Folding patterns are key markers of early brain development and can be studied in vivo with MRI, making them useful for assessing the potential long-term impacts of adverse events on neurocognition. Quantitative characterization of early brain gyrification is therefore essential in the identification of biomarkers linked to neurodevelopmental disorders[20,21], as recent works have shed light on the functional significance linked to subtle variations in folding patterns[18,19,22–25]. Furthermore, a body of literature focusing on premature birth documented the relationship between abnormal brain development and cognitive deficits[14,26–28]. Since gyral patterns stabilize during the postnatal period, early abnormal cortical folding reported in premature-born infants remains consistent, with lasting consequences in frontal and temporal cortices, and overall throughout the brain[14,26–28]. Studies enforce that these early-disrupted gyrification patterns due to preterm conditions are associated with impaired neurobehavioral outcomes in domains such as attention, language, and full scale IQ[26–30].

Accurate and quantitative delineation of gyrification dynamics during the perinatal period, starting in utero up until shortly after birth, is thus of critical importance. Folding trajectories are increasingly discussed in the

[1]Institut de Neurosciences de la Timone, UMR 7289, CNRS, Aix-Marseille Université, Marseille, France. [2]APHM, Service de Neurologie Pédiatrique, Hôpital de la Timone, Aix-Marseille University, Marseille, France. [3]APHM, Service de Gynécologie Obstétrique, Hôpital Nord, Aix-Marseille University, Marseille, France. [4]APHM, Service de Neuroradiologie Diagnostique et Interventionnelle, Hôpital de la Timone 2, Aix-Marseille University, Marseille, France. [5]IMT Atlantique, LaTIM U1101 INSERM, Brest, France. [6]These authors contributed equally: Nadine Girard, Guillaume Auzias. ✉e-mail: angeline.MIHAILOV@univ-amu.fr

literature, with most studies considering fetal and postnatal periods separately, such that the specific impact of birth on gyrification remains entirely overlooked. The impact of the transition from fetal to postnatal life on brain gyrification thus remains to be characterized.

We offer the first quantitative characterization of the impact of birth on brain gyrification using the largest multi-centric perinatal dataset (21–45 weeks post-conception, wPC), including 819 MRI sessions from typically developing fetuses and postnatal participants. We introduce a new unified MRI processing pipeline specifically designed to compensate for uncontrolled variations in cortical measures induced by different acquisition protocols between fetuses and postnatal participants. Through a regression discontinuity statistical approach, we reveal a birth-related discontinuity in the trajectory of gyrification—an effect absent in tissue volume trajectories. Rigorous sensitivity analyses and visual quality assessments were conducted to reduce potential confounding factors. Our findings provide insights into the biological impact of birth on the newborn brain, highlighting influences on gyrification. This key finding warrants consideration in future research on early brain morphology in both typical and atypical neurodevelopmental trajectories.

## Results
### Data selection and description of the final sample
The first challenge to be addressed was the acquisition of a large set of high-quality measures extracted from anatomical MRI data acquired on normally developing fetuses and postnatal participants. In this study, we combined MRI data from two main sources: the developing Human Connectome Project (dHCP) (https://www.developingconnectome.org) that released MRI data from fetuses and postnatal participants (denoted below as Fetal dHCP and Postnatal dHCP), and our local dataset of fetal MRI acquired during routine clinical appointments at la Timone Hospital in Marseille (denoted as MarsFet). See "Methods" below for detailed descriptions of data acquisition and aggregation.

For the **MarsFet** cohort, 149 out of 806 fetal sessions corresponding to "clinical controls with normal MRI" were selected based on neuroradiology clinical evaluations, exclusion criteria, and visual QC of estimated 3D T2w volumes, segmentations, and cortical meshes. For participants from the **Fetal dHCP** dataset, 84 out of the 297 MRI sessions were excluded based on visual QC of the segmentation and cortical mesh, resulting in 213 MRI sessions with accurate morphometric measures. Fetal scans typically result in greater exclusion numbers due to uncontrolled motion in the womb, which is not surprising. To the best of our ability, we included only good quality fetal scans without artefacts in order to limit as much a source of bias as possible. Lastly, for the postnatal participants from the **Postnatal dHCP** cohort, 457 out of 887 MRI scanning sessions were selected based on clinical and imaging exclusion criteria, exclusion of multiple pregnancy and preterm birth, and quality control conducted by the dHCP consortium and in-house[31]. All MRI data were subjected to a unified processing pipeline (essential in obtaining segmentations and cortical meshes for both fetal and postnatal subjects), as well as the same in-house quality assessment procedure. Selection from these three cohorts (MarsFet, fetal dHCP, and postnatal dHCP) resulted in a final normative *perinatal* sample (excluding preterms)

of 819 subjects, making it the largest sample to date, allowing us to assess the neurodevelopment of the perinatal period (Table 1). A flow chart recapitulating the data selection process for healthy participants is provided in the supplementary (Supplementary Fig. 1). To note, the date of conception is estimated using ultrasound in the MarsFet cohort, and the postmenstrual date (i.e., 2 weeks after the last menstrual period) in the fetal dHCP cohort. In this study, however, ages are collectively referred to as wPC.

A preterm subgroup was also included in this study, but not in the estimation and analysis of neurotypical trajectories. As explained below, mapping preterms alongside the neurotypical trajectory of gyrification index computed from non-preterm subjects is informative regarding the respective influence of birth and imaging-related factors, since the preterms were scanned using the exact same MRI protocol as the other postnatal subjects from the postnatal dHCP. The dHCP recorded 305 sessions from 205 babies born before 37 wPC. The application of clinical and imaging exclusion criteria by the dHCP consortium, as well as quality control and exclusion of multiple pregnancy using the same criteria as controls, resulted in a final preterm sample of 102 sessions from 74 preterm babies (Table 1).

### Unified MRI processing pipeline and quality assessment
A unified MRI segmentation and surface extraction pipeline is crucial for quantitatively depicting the development of brain structures during the perinatal period. One key contribution of this work is showcasing the practicability of processing cerebral MRI scans from fetuses and postnatal participants using identical tools, as illustrated in Fig. 1. To achieve this, our approach entails combining a state-of-the-art deep-learning segmentation framework (nnUNet)[32] with a large amount of T2w postnatal MRI data sourced from the dHCP dataset, in order to transfer a segmentation model from postnatal data to fetal imaging. This approach has two key advantages: (1) the segmentation model is trained on top-quality data, which is not accessible to fetal acquisitions since several sources of artifacts affecting the quality of fetal acquisitions can be avoided in postnatal acquisitions; (2) the anatomical nomenclature and segmentation accuracy are unified, meaning that variations in the measures of segmented structures between fetal and postnatal datasets are not impacted by confounding factors related to data processing, contrary to other studies where different segmentation models were used[33]. Each processing step of our approach is described in greater detail in "Methods". Using this procedure, we obtain high-quality measures for the following features: volume of the cerebellum, hippocampi, brainstem, lateral ventricles, external cerebrospinal fluid (eCSF), cortical gray matter (cGM), white matter (WM), and deep gray matter (dGM); as well as the following surface features: the total surface area of the the gray matter/WM interface and the gyrification computed as the ratio between the total surface area and the surface area of the white surface envelope[34]. Analyses in the current study investigated the cerebrospinal fluid (CSF), which is a combination of the eCSF and lateral ventricle volumes (Fig. 1).

### Neurodevelopmental trajectory of the gyrification during the perinatal period
The estimated trajectory for gyrification revealed nonlinear growth and a substantial increase with postconceptional age starting in the second

### Table 1 | Cohort demographic information

| | *n* Sessions (from *n* participants) | Sex ratio (M:F:NA) | Scan age range (postconceptional weeks) | Scanner type ratio (3T:1.5T) |
|---|---|---|---|---|
| Perinatal sample | 819 (*797*) | 424:373:22 | 21.4–44.9 (mean = 36.5; std = 6.1) | 754:65 |
| Fetal Sample I: *MarsFet* | 149 (*130*) | 64:63:22 | 23.7–37.0 (mean = 31.6; std = 2.5) | 84:65 |
| Fetal Sample II: *dHCP* | 213 (*202*) | 112:101:0 | 21.4–36.7 (mean = 29.4; std = 3.5) | 213:0 |
| Postnatal Sample: *dHCP* | 457 (*456*) | 248:209:0 | 37.4–44.9 (mean = 41.5; std = 1.6) | 457:0 |
| Preterm Sample: *dHCP* | 102 (*74*) | 58:44:0 | 26.7–44.1 (mean = 37.1; std = 4.1) Birth age: 24.2–36.8 (mean = 31.8; std = 3.3) | 102:0 |

Basic demographic information for the sessions of the perinatal cohort, which is composed of the MarsFet and dHCP cohorts. Also included are the basic demographic data for the preterm cohort. The perinatal sample is composed of the fetal and the postnatal samples.

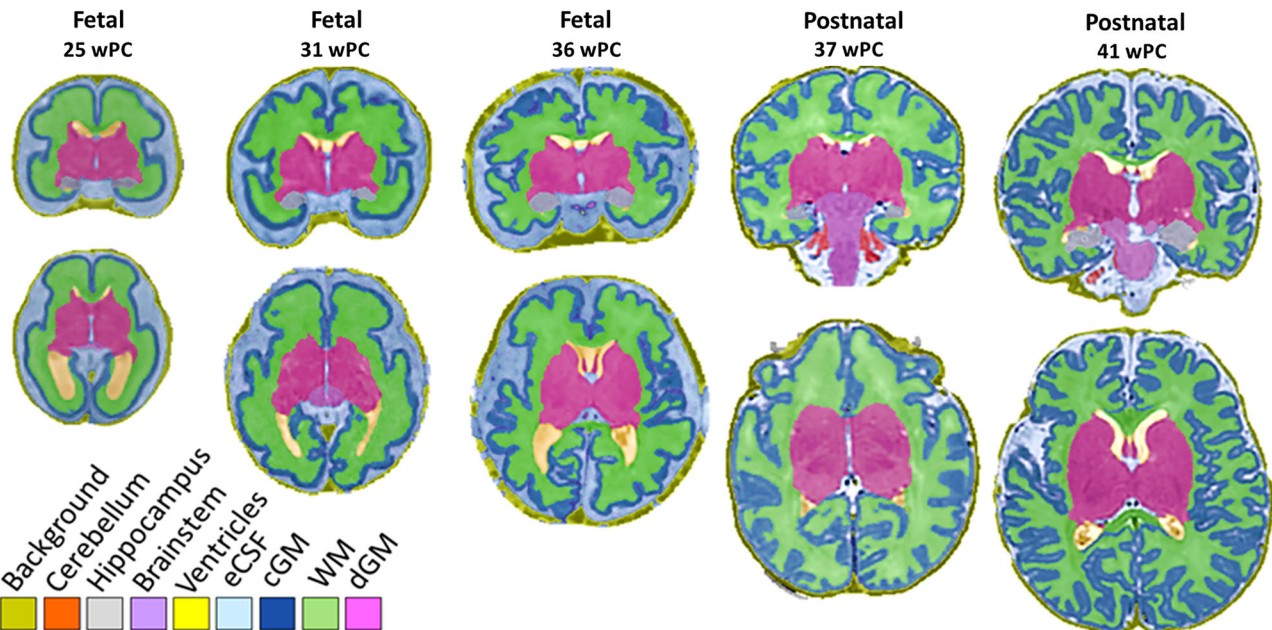

**Fig. 1 | Consistent segmentations across fetal and postnatal participants.** Examples of segmentations for three fetuses and two postnatal participants illustrating the consistency of the anatomical delineation despite large variations in age and developmental stage. eCSF represents external cerebrospinal fluid, cGM represents cortical gray matter, WM represents white matter, dGM represents deep gray matter.

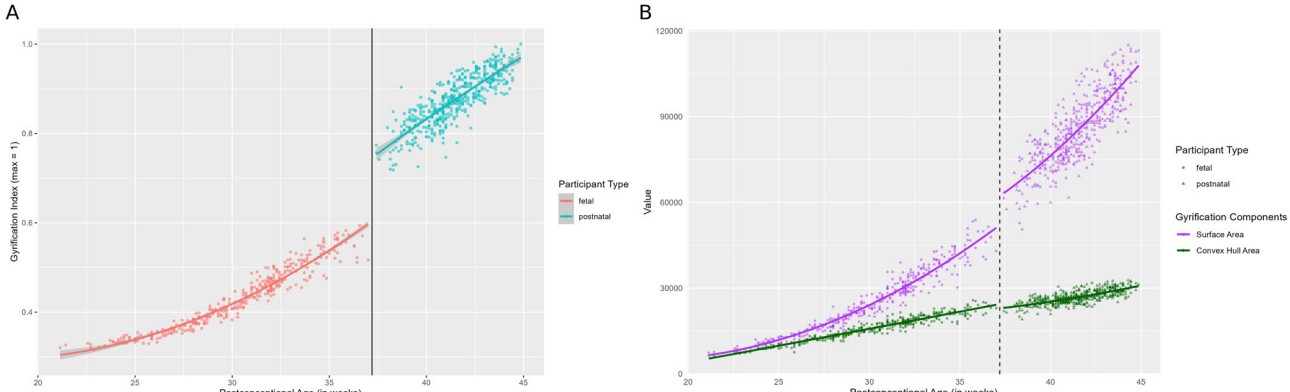

**Fig. 2 | Sharp change at the time of birth across the perinatal trajectory of gyrification and its components.** Plot illustrating the regression discontinuity (effect size = 7.48, $p = 6.44 \times 10^{-14}$) of gyrification in (**A**), and the regression discontinuities of its components (surface area: effect size = 3.68, $p = 2.26 \times 10^{-4}$; and convex hull area: effect size = 1.26, $p = 0.21$) in (**B**) as a function of age at a cut-off of 37 postconceptional weeks (time of birth) ($n = 819$). In (**A**), red dots represent the fetal sample and cyan dots represent the postnatal sample, with a line indicating the time of birth. In (**B**), purple points represent the surface area trajectory, green points represent the convex hull trajectory, dots represent the fetal sample, triangles represent the postnatal sample, and the dotted line indicates the time of birth.

trimester up until almost 2 months of age post-birth. Though the quadratic model fits the data well ($p < 0.001$, $R^2 = 0.96$; residual standard error = 0.04), and thus provides a solid illustration of the neurodevelopmental trajectory of folding during the perinatal period, a noticeable jump can be seen around ~37 postconceptional weeks, which is when birth takes place in our study sample (Supplementary Fig. 3). A quadratic regression analysis was also carried out for the additional features. We report that all brain tissues follow a smooth trajectory as they move from fetal to postnatal periods, without a visible discontinuity seen at the time of birth. Alternatively, we report that though it does not have a sharp change at birth as seen with gyrification, the CSF seems to display a difference in variance and in trajectory before and after birth (Supplementary Fig. 4C). Critically however, gyrification remains the feature for which the sharpest change in trajectory at birth is observed.

To empirically and quantitatively comprehend this noticeable shift in the trajectory of gyrification neurodevelopment within the perinatal period, we further conducted a regression discontinuity design (RDD) analysis.

## Regression discontinuity analysis

An RDD identifies the presence or absence of a discontinuity at a predetermined threshold, or cut-off, on a variable across a regression[35,36]. Using a regression discontinuity model, we show that gyrification increases as a function of postconceptional age beginning in utero and extending into postnatal life, with a prominent jump, or discontinuity, witnessed at the time of birth, and minimal variance across its trajectory (effect size = 7.48, $p = 6.44 \times 10^{-14}$, 95% confidence interval = 0.15, 0.25) (Fig. 2A). The jump in gyrification at birth corresponds to 21.4% of the total growth during the defined perinatal period (21–45 wPC) (computed from the difference between the last fitted value directly before, and the first fitted value directly after the 37-week point). As a matter of comparison, the total growth occurring in the fetal period is 45.0%, and the total growth occurring during the postnatal period is 33.6%. Thus, the jump in gyrification at-term birth represents almost half of the entire growth occurring during the fetal period. Gyrification is a measure composed of the ratio between two

**Fig. 3 | The sharp change in gyrification at birth is illustrated by the relationship between effect size and p value.** Effect size as a function of log-transformed p value from the regression discontinuity analysis in each feature (a higher effect size indicates a larger discontinuity at our cut-off, i.e., at birth at 37 weeks). Gyrification is shown to have a strikingly high effect size compared to other features.

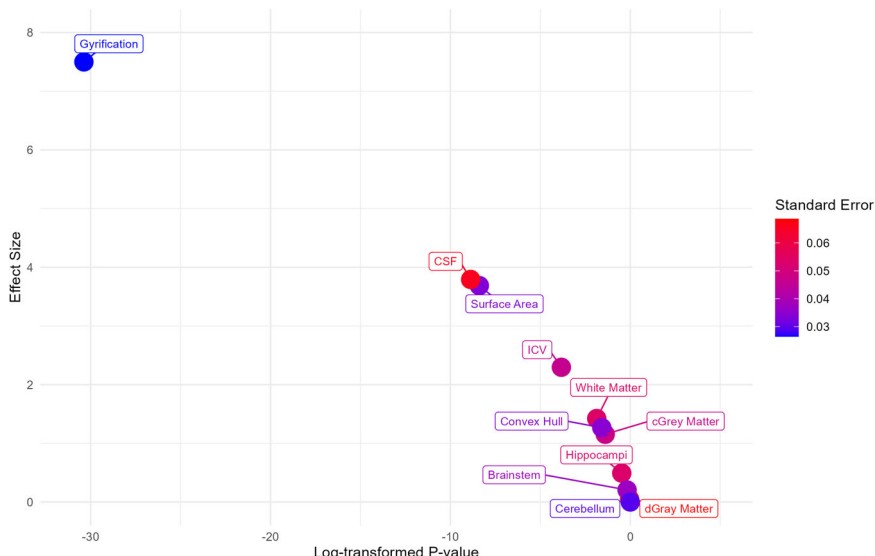

**Fig. 4 | Preterm subjects mapped onto the perinatal trajectory of gyrification.** We show that the neurodevelopment of gyrification as a function of age for preterm participants ($n = 102$) follows a continuous trajectory compared to a jump seen around birth in our typically developing population ($n = 819$).

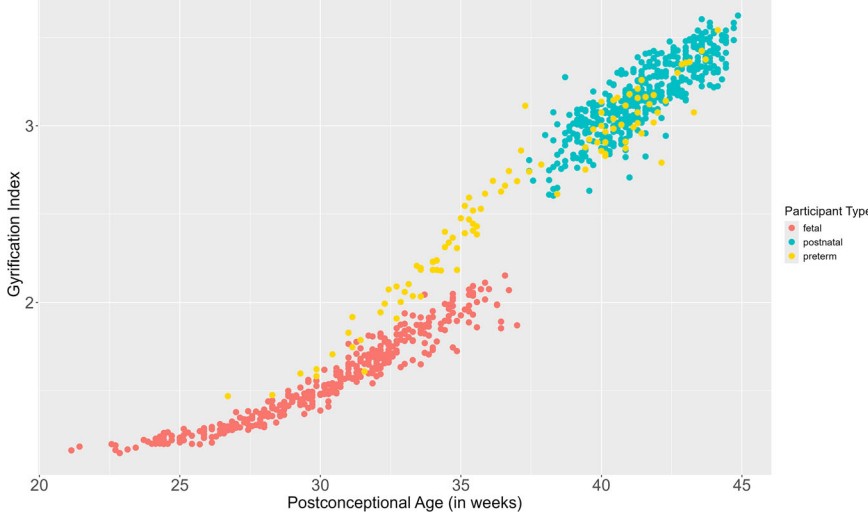

components: the surface area and the convex hull. We decided to also run an RDD on the convex hull in order to help us explain the jump that we are observing in gyrification at birth. The convex hull did not show a high effect size for a discontinuity at birth (effect size = 1.26, $p = 0.21$) (Fig. 2B). The surface area is included here for reference (effect size = 3.68, $p = 2.26 \times 10^{-4}$), though will be separately discussed.

To determine whether this significant jump is specific to gyrification or common to all measures (i.e., to rule out effects being explained by differences in acquisition and processing before and after birth), we also ran an RDD analysis on all brain tissue volumes, surface area, and the CSF. The effect size as a function of log $p$ value for all measures is reported in Fig. 3. We did not observe any discontinuities at birth for cGM (effect size = 1.15, $p = 0.24$), subcortical gray matter (effect size = 0.04, $p = 0.96$), WM (effect size = 1.42, $p = 0.15$), hippocampal (effect size = 0.49, $p = 0.62$), brainstem (effect size = 0.20, $p = 0.84$), cerebellar (effect size = $4.94 \times 10^{-6}$, $p = 0.99$) and the overall intracranial volume (effect size = 2.29, $p = 0.22$) (Fig. 3, Supplementary Fig. 7). All significant $p$ values are Bonferroni corrected. We observed larger effect sizes for surface area (effect size = 3.68, $p = 2.26 \times 10^{-4}$) and CSF (effect size = 3.79, $p = 1.49 \times 10^{-3}$), indicating a stronger effect of birth on these measures compared to tissue volumes. Nonetheless, these effect sizes remain much lower than that of gyrification

(effect size = 7.49, $p = 6.44 \times 10^{-14}$). The clear continuity in the trajectory along the perinatal period in all brain tissues, both visually (quadratic models) and statistically (regression discontinuity models), collectively confirms the effectiveness of our unified image processing pipeline in compensating for variations in MRI acquisition settings between fetuses and postnatal participants. This supports the assumption that the observed jump at birth is not due to inconsistencies in image processing. It is unlikely that such bias would specifically target surface area and convex hull measures and thus induce a much higher effect size compared to all other measures submitted to the same image processing pipeline.

### Sensitivity analyses to strengthen the biological relevance of our results

**Acquisition settings.** Since it is not possible to use identical acquisition conditions for fetuses and postnatal subjects, we cannot fully eliminate possible contamination by factors related to MRI acquisition settings. One way to address this potential concern, in addition to our RDD analysis and thorough visual quality checks, is by mapping preterm-born subjects (born before 37 wPC, but scanned using the exact same protocol as the postnatal dHCP and the same scanner as Fetal dHCP) across the typical neurodevelopmental trajectory of gyrification (spanning from 21 to 45 wPC). We show in Fig. 4

how preterm subjects start with similar values in gyrification as for typically developing fetuses, and then progressively diverge in development as they approach birth. Then, from ~37 to 38 weeks on, preterm participants map directly along the typically developing trajectory of postnatal participants. This shows that preterm data increase as a function of age, forming a smooth transition between the typically developing fetuses and postnatal participants. The congruence between preterm and fetal data supports the effectiveness of our image processing pipeline in helping compensate for variations in MRI acquisition. The continuity observed for preterms but not for typically developing participants confirms that the discontinuity we observe in gyrification is likely not due to differences between MRI acquisitions (2D sequence inside the womb for fetuses and 3D acquisitions outside the womb for postnatal participants) or image processing, but to biological and/or mechanical processes acting at the time around birth.

**Scanner effects in fetal data.** Since our fetal cohort contains three different scanner types, we assessed potential scanner effects in gyrification using regression models. An ANOVA indicates that there was a significant effect of scanner type within the fetal sample ($p = 4.29 \times 10^{-4}$). This effect stemmed from differences between each 3 T scanner (of the MarsFet and of the dHCP cohort) and the 1.5 T scanner (of the MarsFet cohort). For the MarsFet 3 T scanner versus the MarsFet 1.5 T scanner—effect size = 0.24, $p = 0.015$. For the dHCP 3 T scanner versus the MarsFet 1.5 T scanner—effect size = 0.09, $p = 8.35 \times 10^{-5}$. No significant difference between the two 3 T scanners was reported (effect size = 0.15, $p = 0.255$). Statistical differences between scanner strengths are likely attributed to residual influences of the acquisition sequence and settings on the estimated surfaces, which is to be expected. What we are interested in confirming, however, is that the trajectory pattern of gyrification as a function of age is the same across all scanner types, which was indeed still the case (Supplementary Fig. 8).

**Sex effects.** We did not stratify our cohort by sex when looking at the gyrification trajectory due to incomplete sex data. However, to better understand its role, we selectively analyzed participants with sex data ($n = 797$, 424 males and 373 females) to determine potential sex-specific effects. Upon including sex as a covariate in the RDD model, discontinuity at birth was still statistically significant, and the effect size remained high with minimal change to the confidence interval (effect size = 7.58, $p = 3.28 \times 10^{-14}$, 95% confidence interval = 0.14, 0.23). Upon running RDD models independently in males and females, we found yet again that our overall results were maintained and that there was still a discontinuity in the developmental trajectory at birth (for males: effect size = 2.83, $p = 4.57 \times 10^{-3}$, 95% confidence interval = 0.80–0.41; and for females: effect size = 6.39, $p = 1.63 \times 10^{-10}$, 95% confidence interval = 0.19–0.36)(Supplementary Fig. 9).

**Variations in image processing.** We report in (Supplementary Note 1, Supplementary Figs. 12 and 13) a detailed comparison between BOUNTI and our technique based on nnUNet, showing that the accuracy was highly similar for fetuses, but noticeable differences were present for postnatal participants. Importantly, the overall quality was sufficient to demonstrate the robustness of the discontinuity of gyrification with respect to the segmentation approach. With BOUNTI, we again observe a noticeable jump of gyrification after birth, and therefore a very high effect size (effect size = 7.16, $p$ value = $6.23 \times 10^{-12}$; 95% confidence interval = 0.16, 0.27) (Supplementary Fig. 10A). We also still observe a relatively high effect size for the CSF, and continuous trajectories for all other measures (Supplementary Figs. 10 and 11, Supplementary Table 2).

## Discussion

Shifting from an intra- to an extrauterine environment requires adaptation that affects the entire fetal/newborn body, including the brain. To the best of our knowledge, this is the first neurodevelopmental study targeting the age-related trajectory of gyrification in perinatal subjects crossing birth that includes MRIs from both fetal and postnatal subjects, with homogenized preprocessing techniques and quality assessment. As confirmed by the literature, we observe that global gyrification drastically increases in utero and continues to do so postnatally[9,18,37]. With respect to the overall perinatal period investigated in this study, we report a substantial jump of 21.4% in gyrification around the time of at-term birth. This is equivalent to almost half of the total increase occurring during the fetal period (a total growth of 45.0%) and roughly two-thirds of the total increase occurring during the postnatal period (a total growth of 33.6%).

To better interpret the observed discontinuity of gyrification, we further estimated effects at birth using an RDD in additional traits, including surface area and volumetric features in order to enforce the fact that the change seen in gyrification at birth is accurate and specific to this measure. Though two of these additional features showed a significant discontinuity at birth (surface area and CSF volume), we observe a clear continuity for the remaining tissue volumes. This reveals that not all cortical features behave in the same way as the brain moves from a prenatal to a postnatal environment. Furthermore, this demonstrates that our unified segmentation and preprocessing protocol between fetuses and postnatal participants accurately depicts the effect of birth on the brain. Otherwise, if methodological biases were responsible for the discontinuity observed in gyrification, these biases would likely have introduced some form of discontinuity in *all* mapped features (in addition to gyrification).

The gyrification trajectory of preterm participants was also mapped alongside that of our typically developing sample. In the early fetal period, we illustrate how preterm participants' level of folding is close to that of typically developing subjects, and how, as age progresses, it begins increasing and diverging. Then, as we surpass birth, subjects begin mapping directly onto the trajectory of the typically developing postnatal sample. Most studies investigate later behavioral, cognitive, or morphological outcomes as a consequence of premature birth, though extremely few look at the direct difference in morphology between preterms and age-equivalent fetuses, and between postnatal participants and age-equivalent preterms, early on in life. In a single study, we successfully illustrate how preterm participants can match the gyrification trajectory of typically developing age-equivalent participants at only certain periods. The pattern of similarity, divergence, and later convergence in preterm subjects suggests that the discontinuity in typical-developing gyrification is biologically and/or mechanically driven by birth, not differences in acquisition settings.

These relevant findings illustrate how gyrification neurodevelopment differs between typical and preterm participants around birth, and across the perinatal period. Revealing this distinction is important since altered gyrification in preterms is traditionally associated with later cognitive and behavioral outcomes[27,38–41]. Furthermore, we confirm the preliminary results of Lefèvre et al., that report pronounced folding in premature infants compared to fetuses at equivalent ages[14]. Those results, coupled with the continuous gyrification neurodevelopment in preterms reported in the present study, both enforce the idea that the event of birth itself (and not necessarily the age) is linked to a folding discontinuity at-term birth in typically developing participants. Interestingly, though early regional differences in gyrification between preterms and typically developing newborns have been reported in the literature[30,42–44], we show that the level of *global* gyrification remains the same between the two groups during the first months of the postnatal period. Since gyrification is a strong indicator of behavioral and cognitive outcomes, further study is required to determine how the difference in neurotrajectory around birth between preterm and typically developing individuals is linked to later neurodevelopment.

Another relevant approach to enforce the biological validity of this jump in gyrification at birth is to apply an alternative segmentation technique. We did this using the BOUNTI segmentation protocol, which reaffirmed a burst in gyrification. Similarly, the CSF also showed marked changes after birth, while remaining tissue volumes showed continuity and low effect sizes as they traverse birth, exactly as shown upon applying our in-house nnUNet segmentation technique.

Gyrification is known to be a complex combination of genetic, mechanical, and external constraints that begins forming early in the gestational period[9,45,46]. A putative role of its development is to draw regions of connectivity closer to one another to decrease action potential transit time and, in turn, increase overall brain internal communication efficiency[47–49]. The jump in gyrification that we report after birth (Fig. 2A) is consistent with qualitative radiological observations[50–52], and could be linked to the crucial increase in brain activation after birth attributed to abrupt and intense sensory stimulation from exposure to the extrauterine environment[53,54]. Such observations could also be consistent with the evolution of tertiary gyri, which are the final set of gyri that begin forming around the same time[22]. Our results suggest that tertiary gyri may be directly affected by birth-related factors[55].

Furthermore, since the surface area component of gyrification accelerates faster than the convex hull in growth as it approaches birth, biomechanical factors may also be at play, revealing underlying dynamics of gyrification. Compared to the convex hull area, faster surface area acceleration likely plays a bigger role in constraining the brain into a restricted space. Additionally, during birth, molding temporarily compresses certain areas of the skull via intense pressure from passing through the mother's birth canal. Within a few weeks, the brain rounds out in healthy subjects[56]. Meanwhile, the early neonatal period is a dynamic period for rapid gyrification expansion and development to ensure normal growth and therefore normal cognitive and motor development[18]. The event of molding, in addition to several other biological and environmental factors, could help contribute to a plausible interpretation of why we see changes in gyrification after birth[18,56].

Finally, note that in the present work, we chose to use an RDD model that assesses potential discontinuity in the measures because it is well fitted to the sampling in time that is imposed by the data. Since the timing of fetal and postnatal acquisitions was not controlled with the aim of having high sampling close to birth in the present dataset, we cannot accurately estimate speed and acceleration close to birth. Future work will be required to closely investigate the continuity versus discontinuity aspects of the sharp transition we report in the present study.

In addition to gyrification, we report two other features that exhibit a significant change in trajectory at birth: the surface area and the CSF. The relationship between gyrification and surface area is well established after birth, as they are positively correlated and genetically related[45,48,57,58]. Yet the relationship between gyrification and the CSF volume is slightly more complex. We report a high variance in CSF volume during the fetal phase, followed by a notable drop in variability during the postnatal phase, which reflects many qualitative observations[59–61]. The CSF partakes in a plethora of roles, including mechanical and immunological protection, homeostasis, delivery of neural growth signaling molecules, neurotoxic waste elimination, and regulation of brain growth via positive hydrostatic pressure[62–67]. Furthermore, CSF constituents are known to vary according to age, from the embryonic phase up until adulthood, and experience an abrupt drop in protein concentration after birth[68,69].

Another factor to be considered is that the mature brain is the result of a multifaceted array of neuro-ontogenetic processes beginning shortly after conception, including neurulation, neurogenesis, synaptogenesis, pruning, myelination, and neuronal migration[3,70,71]. Neuronal migration, in particular, is the process in which neurons travel along the cortex to reach their final destination in the brain, reportedly guided by CSF flow, and typically halts and/or slows down at birth[3,72,73]. The completion of neuronal migration has been linked to stabilized gyrification, as corroborated by biomechanical and animal models[74–76], while disruption of this process leads to *neuronal migration disorders* that are largely based on folding malformations[77].

This shift in gyrification after birth can moreover be partially explained by the mechanical influence of the CSF on the two-layer (gray matter and WM) structure of the cerebral cortex[78]. Inside the skull, the brain is subject to pressure from the CSF, which, when abnormal (i.e., too high or too low), can cause variations in gyrification[79,80]. Patients with hydrocephalus illustrate this by exhibiting flattened gyri and narrower sulci, likely linked to higher volumes and therefore a higher pressure from the CSF[78,81]. Developing newborns lose ~5–7% of their weight in fluids when born, including a decrease in CSF[82,83]. We report this same observation (Supplementary Fig. 7C), which could be linked to a decrease in brain pressure following birth.

The multi-functionality of the CSF, coupled with the robust genetic impact on gyrification, underscores the importance of their early quantitative characterization. The fact that these cortical traits are heavily impacted at the time of birth indicates possible physiological and/or mechanical influences linked to the womb, the maternal environment, or the sudden external environment. Further study is warranted into this complex relationship and how the role of the CSF could be a potentially new component to be taken into account in the development of biomechanical models of the brain.

The current study holds many strengths. First, we combine fetal and postnatal participants, filling a gap in the literature where these subgroups are often studied independently, using the largest perinatal sample known to date. Second, our study benefited from the application of specifically designed homogenized image processing tools based on state-of-the-art techniques, allowing us to accurately analyze brain features from different acquisition protocols, which has never been done before. Lastly, we ran an RDD on the age-related perinatal changes of global gyrification to statistically characterize the presence of nonlinear and abrupt changes in cortical development. This is a novel approach in the field of neuroimaging and provides insights into the impact of birth timing on brain development.

Nonetheless, this study is not without limitations. Though we do our best to accommodate any biases that may be introduced due to unavoidable differences in subject environments (fetal versus postnatal) via the homogenization of image processing tools, we cannot entirely eliminate all possible effects since these populations will always be measured in different environments. Specifically, the proposed unified image processing approach can mitigate but not fully compensate for the confounding effects of image acquisition, cohort, or age. There is simply no way to disentangle confounding effects related to in utero versus ex-utero MRI acquisition. Furthermore, it is important to note that any remaining subtle motion not directly visible by the human eye, termed as "micro-motion" in Alexander-Bloch et al., might affect our results[84].

Another limitation is that we focused on a birth age cut-off of 37 wPC since this is the threshold for being considered at-term based on references from the World Health Organization[2]. Moreover, this study did not include fetal participants older than 37 weeks due to several challenges. These challenges include limited access, primarily due to ethical considerations, as well as the exclusion of the few available older participant scans due to lower image quality. Future studies involving sufficient data points representing the weeks closest to birth to allow for a direct comparison between fetal and postnatal subjects at the same postconceptional ages are necessary to strengthen or disprove the hypotheses presented in the current paper. Despite these limitations, setting the cut-off at 37 wPC still enabled us to confirm the impact of birth on brain anatomy in a typically developing population. As more data become available, it would be valuable to extend the analysis to include older at-term age cut-offs and potentially even non-term age cut-offs using RDD analyses. Also, participants having both fetal and postnatal data were limited ($n = 52$), prompting us to apply a cross-sectional design, which may be thought of as a drawback in the interpretation of brain neurodevelopment. However, in the case of the present study, note that even a longitudinal design would not be sufficient to rule out acquisition-related confounding factors since it is impossible to use the exact same setup (for example, using a head coil for postnatal versus a heart coil for fetal participants during MRI acquisition). Therefore, conducting a longitudinal design could definitely be an advantage, but it would not solve the dilemma of different acquisitions, or of disparate physical environments between fetal and postnatal participants (Supplementary Note 2, Supplementary Fig. 14). Moreover, obtaining a sufficient amount of longitudinal data, with a sufficient amount of time points, is an extremely challenging feat that will hopefully be achieved in the years to come. Another limitation is

**Table 2 | MRI acquisition settings for the MarsFet dataset**

| Scanner model | Skyra | | | | SymphonyTim | | |
|---|---|---|---|---|---|---|---|
| Magnetic field strength (Tesla) | 3.0 | | | | 1.5 | | |
| Resolution (mm³) | 0.6 × 0.6 × 3.0 | 0.7 × 0.7 × 3.0 | 0.8 × 0.8 × 3.0 | 0.7 × 0.7 × 3.6 | 0.7 × 0.7 × 3.0 | 0.7 × 0.7 × 3.4 | 0.7 × 0.7 × 3.5 |
| Repetition Time (Tr) (ms) | 4060 ± (609) | 3418 ± (922) | 3315 ± (911) | 3550 ± (71) | 1740 ± (0) | 1690 ± (0) | 1689 ± (59) |
| Echo Time (Te) (ms) | 180 ± (1) | 177 ± (0) | 177 ± (1) | 177 ± (1) | 141 ± (0) | 138 ± (0) | 131 ± (1) |

A detailed table summarizing MRI acquisition settings for the fetal MarsFet dataset.

**Table 3 | Exclusion criteria for the MarsFet dataset**

- Multiple pregnancy
- Brain malformation: microcephaly (−3SD), macrocephaly (+2SD), commissural anomalies (corpus callosum anomalies, septal agenesis), posterior fossa anomalies (Dandy–Walker spectrum, compressive arachnoid cysts, mega grande cisterna > 12 mm, cerebellar hypoplasia, pontocerebellar malformations), clastic lesions, cortical developmental anomalies
- Enlargement of ventricles superior than 10 mm at ultrasound
- Extracerebral malformation (particularly cardiac or syndromic)
- Genetic syndrome and/or chromosomal abnormality and/or deleterious mutation
- Fetal alcoholism
- Detrimental perinatal event: difficult traumatic delivery, anoxo-ischemia, respiratory distress, convulsion, maladjustment to extrauterine life
- Diabetic mother (treated) or gestational diabetes
- Epileptic mother (treated)

Fetal MRI sessions in the following situations were not excluded:

- Maternal infection by cytomegalovirus, for which the infection of the fetus was ruled out by test on amniotic fluid or urine in the neonatal period
- Presence of small (inferior to 3 mm) periventricular cysts

Detailed inclusion/exclusion criteria for the fetal MarsFet dataset.

that subjects were labeled as typically developing through the inclusion of normal-looking scans, absence of maternal medication, no history of illness, and no reported perinatal complications. A more accurate categorization, however, could have been obtained using neurobehavioral assessments starting at 3–5 years of age, which are unfortunately not yet available for the current study. Nonetheless, the small variance in our measures with respect to the large size of our sample confirms that our results are representative of a normal population.

Gyrification is a key feature established early in life with strong genetic influence, making it an excellent biomarker candidate, with numerous studies linking cortical folding patterns to later cognitive outcomes[20,58,85–87]. By integrating data from typically developing fetal and postnatal subjects, our study bridges a crucial gap by examining age-related changes in gyrification during the perinatal period, focusing on the impact of birth. We proposed a unified segmentation and preprocessing technique across all subjects to help mitigate inevitable biases related to MRI acquisition. We find that while gyrification grows steadily during fetal and postnatal stages, a sharp surge occurs shortly after birth, marking a critical developmental transition. This sudden change likely stems from mechanical and/or physiological adjustments during the transition from an intrauterine to an extrauterine environment, warranting further research into its long-term neurocognitive and neurobehavioral effects.

## Methods

In this section, we outline the datasets and pipelines specifically designed for processing MRI data acquired in both fetuses and newborns. Extensive quality assessments of each step were implemented to obtain accurate, high-quality quantitative measures of brain morphology.

### Fetal dataset I: MarsFet

We constituted our fetal dataset by retrospective access to MRI data acquired during routine clinical appointments at la Timone Hospital in Marseille between 2008 and 2021. Over this period, more than 800 cerebral MRI sessions, spanning a developmental period of 20.3–37.1 wPC, were administered with informed consent following an indication requested by the Multidisciplinary Center for Prenatal Diagnosis, in the context of usual obstetric assessment during pregnancy. This study was approved by the local ethical committee from Aix-Marseille University (N°2022-04-14-003). We focused on the cerebral MRI sessions acquired using a T2-weighted (T2w) half-Fourier single-shot turbo spin echo (HASTE) sequence on two MRI Siemens scanners (Skyra 3 T and SymphonyTim 1.5 T). The details of the MRI acquisition settings are provided in Table 2. To define our fetal normative cohort, close collaboration between contributing medical doctors enabled the design of a set of exclusion criteria combining neuroradiology, obstetrics, and pediatric neurology aspects (Table 3).

Fetal MRI acquisitions consisted of several 2D T2w images with varying acquisition directions (such as coronal, sagittal, axial, or transverse), referred to as *stacks*. Fetal MRI stacks were denoised using the non-local means approach[88] implemented in the ANTS software[89]. A mask of the brain was computed for each 2D image using Monaifbs[90]. The denoised MRI stacks were corrected for bias related to magnetic field inhomogeneity using the N4 method[91] implemented in ANTS. A 3D high-resolution T2w MRI volume (0.5 mm iso) was then estimated from the pre-processed MRI stacks using NESVOR v0.2[92] (https://github.com/daviddmc/NeSVoR). The quality of the resulting 3D high-resolution T2w volume can be affected by various inaccuracies or artifacts related to the quality of the initial stacks. Since the 3D volume quality in turn impacts the accuracy of the segmentation, we visually inspected each reconstructed volume and excluded data of insufficient quality.

The addition of the MarsFet cohort in this study is instrumental since it significantly increases our sample size, which is essential in improving the balance between fetal and postnatal data in the RDD models. It also allows us to assess the influence of acquisition settings on the features extracted from fetuses, since the variations in acquisition settings between the MarsFet and dHCP include critical variations in the way the MRI is acquired. Specifically, MarsFet scans are acquired in clinical settings, while the dHCP

scans are acquired in research-oriented settings. Thus, what is labeled as "variations in acquisition settings" goes beyond simple variations in MRI sequences.

### Fetal dataset II—fetal dHCP

We used the fetal MRI data from the fourth release of the publicly available dHCP (https://www.developingconnectome.org). This dataset consists of 297 MRI sessions from 273 fetuses. As detailed in (https://cds.ismrm.org/protected/19MProceedings/PDFfiles/0244.html), MRI data were acquired on a 3 T Philips Achieva scanner using a 32-channel cardiac coil. Structural T2w data were acquired from six uniquely oriented stacks centered on the fetal brain using a zoomed multiband single-shot TSE sequence, at an in-plane resolution of $1.1 \times 1.1$ mm$^2$ and 2.2 mm slices. We relied on the 3D isotropic (0.5 mm iso) MRI scans reconstructed using the Cordero-Grande et al. method as provided by the dHCP[93].

### The postnatal dataset—neonatal dHCP

We used the third release of the publicly available dHCP neonatal dataset (https://www.developingconnectome.org) that consists of 887 MRI scans from 783 infants, spanning a developmental period of 26.7–44.9 wPC[31]. This dataset contains the data from babies born preterm (before 37 wPC). As detailed in ref. 31, MRI data were acquired on a 3 T Philips Achieva scanner using a neonatal 32-channel phased array head coil, differing from fetal subjects that require the use of a cardiac coil. T2-weighted (T2w) multi-slice fast spin echo scans were acquired at an in-plane resolution of $0.8 \times 0.8$ mm$^2$ and 1.6 mm slices. We relied on the 3D isotropic (0.5 mm iso) MRI scans reconstructed using the Cordero-Grande et al.[93] method as provided by the dHCP. For our normative postnatal cohort, we excluded MRI scans from participants born prematurely before 37 wPC, exhibited clinical anomalies, presented incidental findings that were likely to affect further processing (dHCP radiological score of 3, 4, or 5), and/or were part of a multiple pregnancy. MRIs from postnatal preterm subjects born before 37 wPC were considered as a separate subgroup.

### Image processing steps

#### Training of the model and segmentation on the postnatal dataset.
The dHCP consortium provides whole-brain multi-tissue segmentation maps based on the technique Draw-EM[94] that combines a spatial prior and a model of the intensity of the image in order to enforce robustness to variations in image intensity distribution related to brain maturation. This segmentation map includes eCSF, cGM, WM, lateral ventricles, dGM, cerebellum, brainstem, and hippocampus. To avoid the influence of local inaccuracies in the segmentation of the convoluted cGM resulting from the well-known limitations of atlas-registration approaches (including Draw-EM), we replaced the initial cGM label with the cortical ribbon volumetric mask also provided by the dHCP and defined as the voxels located between the inner (white) and the outer (pial) cortical surfaces. We then use the segmentation maps as defined above from the 50 young infants of the dHCP to train a first nnUNet model. These 50 preterm participants have a scan age range of 29.3–37.1 wPC (34 males, 16 females), and a birth age range of 25.6–36.9 wPC (mean age = 32.7, std = 3.1), covering a substantial portion of the fetal period. Conceptually, the preterm brain can be seen as an intermediate level of maturation between the fetal and postnatal states. By designing a first model on preterm data that is then extended to fetuses and postnatal participants using fine-tuning, we are able to transfer across developmental periods the information carried by the anatomical nomenclature underlying the segmentation process. We denote this model as Unet-preterm. We then applied the Unet-preterm model to predict the segmentation of the remaining subjects from the postnatal dHCP dataset. The visual assessment described in "Quality control (QC)" below confirmed that this segmentation model is more accurate than Draw-EM provided by the dHCP consortium.

#### Fine-tuning and segmentation of the fetal dataset.
We used the Unet-preterm model described above to predict the segmentation from the 3D

T2w fetal MRI volumes that passed the visual quality assessment. As expected, the accuracy of these segmentations was not always satisfying. We adopted the "active learning" strategy as introduced in refs. 95,96 by manually refining the initial segmentation predicted by the model before running again the fine-tuning procedure. We specifically screened the highest quality fetal MRI volumes, which are critical for accurate manual delineation of anatomical structures, and selected seven cases showing minimal errors that could be corrected without ambiguity. We obtained optimal quality data and corresponding accurate segmentation for these seven fetuses. We then fine-tuned the Unet-preterm model using these ground truth fetal data following the procedure described in (https://github.com/MIC-DKFZ/nnUNet/blob/master/documentation/pretraining_and_finetuning.md). During this fine-tuning, the model further converges toward an optimal configuration that lies closer to the Unet-preterm model than in the case of training directly from fetal data. This second model is denoted as Unet-fetus. Extensive visual assessment of selected cases for which the prediction from the Unet-preterm was inaccurate confirmed that the fine-tuning on seven fetal data points was sufficient to proceed with the next step of our pipeline.

#### Whole-brain mesh generation and surface features.
In order to extract a surface mesh allowing us to compute surface features, namely gyrification and surface area, we designed an extended WM mask from the multi-tissue segmentation by aggregating the labels corresponding to the ventricles, dGM, and hippocampus with the WM. Supplementary Fig. 2 shows examples of cortical surfaces extracted using our pipeline spanning the perinatal period from fetal to postnatal subjects. A 3D global gyrification index similar to the one proposed in ref. 97 was computed as the ratio between the white mesh surface area and the area of its convex hull derived directly from the mesh using the SLAM toolbox (brain-slam/slam: Surface anaLysis And Modeling).

#### Quality control (QC).
In order to ensure a high accuracy of the features used in our statistical analyses described below, we conducted an extensive visual assessment of the quality of the images, segmentations, and corresponding cortical surfaces. Three raters (G.A., A.M., and A.P.) scrutinized the 3D reconstructed T2w volumes to identify potential artifacts and anatomical inaccuracies. We also visually controlled the quality of the cortical surface mesh for each participant from all three datasets: dHCP postnatal, dHCP fetal, and MarsFet fetal. The visual assessment of the cortical surface is efficient for spotting subtle segmentation inaccuracies that would be very hard to detect by inspecting the segmentation in the voxel space.

### Statistical analyses

#### Neurotypical trajectory of gyrification.
We estimated the neurodevelopmental trajectory of gyrification in control fetuses and postnatal participants, during the perinatal period from 21 to 45 wPC. This estimation was computed including both fetal and postnatal subjects from our perinatal cohort, but after exclusion of the preterm subgroup since they cannot be considered as controls (as previously described). We applied a quadratic regression model as proposed in refs. 12,98. The gyrification trajectory showed a prominent jump, or discontinuity, at the time of birth, thus prompting us to conduct an RDD to properly quantify this pattern.

#### Regression discontinuity design.
As described in Calonico et al., RDD is used to estimate the causal effect of an intervention[35]. This type of design was introduced over 60 years ago, however, only recently gained vast popularity in the medical field[35,99–101]. This method identifies the presence or absence of a discontinuity at a predetermined threshold, or cut-off, on a variable across a regression[35,36]. Specifically, observations directly below and above the cut-off are assumed to be similar in all respects except for the condition set at the cut-off (in our case, birth). By considering only a focal set of data directly above and below this cut-off,

we can better explain the causal effect of an event such as birth on brain morphology, and further minimize influence from confounding factors along a trajectory. Its computation is therefore local in nature and is run using local polynomial nonparametric regression with statistical inference based on large sample approximations. It is important to note that an RDD model does not estimate the cut-off itself but rather requires a predetermined cut-off value to be fed into the model, thus testing a hypothesis of discontinuity at a point of interest across a continuous variable.

The use of an RDD in the context of structural neuroimaging along a trajectory brings originality to our study and to a field where this type of analysis is novel. Neurodevelopmental neuroimaging studies often seek to understand how cortical structures, such as gyrification, develop across different stages of life, with most studies traditionally utilizing longitudinal or cross-sectional designs. However, analyzing how brain features evolve with age around notable life events such as birth is crucial in providing insights into the developing brain and therefore needs to be better characterized. Using a regression discontinuity analysis at the time of birth thus provides an original approach in examining how this event impacts gyrification during a period where gyrification is being established.

The rdrobust function in R was used to implement an RDD on gyrification neurodevelopment, which adopts a bias-corrected inference approach computing the effect size, statistical significance, and confidence intervals at a set cut-off, which in our case is at birth[35]. Specifically, the effect size was computed by dividing the standard error by the coefficient, which are both outputs from the rdrobust regression discontinuity model. Our cut-off was set at a postconceptional age of 37 weeks since we are interested in studying a typically developing population, and 37 weeks is defined as an at-term, and therefore non-premature, birth[2,102,103]. The following formula is used for the left side of the cut-off, i.e., the fetal sample (before birth):

$$Y_i = \beta_0 + \beta_1(X_i - c) + \beta_2(X_i - c)^2 + \ldots + \beta_p(X_i - c)^p + \varepsilon_i \, for \, X_i < c$$

While the next formula is used for the local polynomial regression on the right side of the cut-off, i.e., the postnatal sample (after birth):

$$Y_i = \gamma_0 + \gamma_1(X_i - c) + \gamma_2(X_i - c)^2 + \ldots + \gamma_p(X_i - c)^p + \varepsilon_i \, for \, X_i \geq c$$

These formulas summarize an RDD model computation for a given outcome $Y$—in this case, the standardized gyrification index (set at maximum = 1), a running variable $X$—in this case, postconceptional age, and a cut-off $c$—set at birth during the 37th wPC. Variable $p$ represents the local polynomial regression order, $\beta$ and $\gamma$ represent the coefficients for either side of the cut-off $c$, and $\varepsilon$ represents the error.

Furthermore, we computed the percent increase at birth from the difference between the last fitted value of the local polynomial regression before birth, and the first fitted value of the local polynomial regression after birth (relative to the entire perinatal period).

To properly compute an RDD model, meta-parameters need to be defined a priori, including the selection of polynomial order, the type of kernel, and the bandwidth around the cut-off. To reduce bias and to capture non-linearity that is typically present in neurodevelopmental modeling, a second-order polynomial quadratic model was selected to provide accurate estimates of the effect size around the cut-off[12,33,74,104,105]. Regarding kernel selection in the model, a triangular kernel was deemed appropriate since it places more weight on observations around the cut-off, which are most informative in estimating effect size[36,106]. This aids in bias reduction since points further away from the cut-off are not likely to reflect true discontinuity but rather are more likely to introduce undesired noise into the estimation. Lastly, to obtain a final bandwidth *length*, it is necessary to choose the appropriate bandwidth *selection method*, which will ultimately guide the model into estimating the final bandwidth *length*, which represents the amount of observations to be included on either side of the cut-off[107]. We provide an extensive description of our parameter model selection process in (Supplementary Note 3, Supplementary Figs. 5 and 6, Supplementary Table 1).

**Neurodevelopmental trajectory of classical cortical features.** It is critical to carefully assess whether the discontinuity in the trajectory of the gyrification index could be related to a non-biological confounding factor, for example, differences in the advanced image processing pipeline. To evaluate this, a first step consists of conducting RDD analyses on the other features extracted using the exact same image processing pipeline as for the gyrification index: cortical surface area and brain volumes including gray matter, subcortical gray matter, WM, hippocampal, brainstem, cerebellar, CSF (a combination of the eCSF and the ventricles), and overall intracranial (which is computed as a combination of all volumetric features).

**Sensitivity analyses.** To further assess the robustness of our statistical models, we tested for sources of potential influence on the estimated trajectories by comparing different populations using identical acquisition settings and protocols, by investigating the effect of sex and of the type of scanner used for MRI data acquisition, and by comparing different segmentation methods.

Scanner, acquisition settings, and image processing. Since this study combines fetal and postnatal participants that cannot be scanned using identical acquisition settings and protocols, careful assessment of the potential influence of these factors on our results is critical. For this reason, we minimize potential effects by using the same segmentation tools and by conducting thorough visual quality checks. However, this still does not fully eliminate potential contamination by factors related to MRI acquisition settings. To fully address this potential concern to the best of our ability, we mapped dHCP preterm participants alongside the typically developing gyrification trajectory. Since these participants are from the dHCP cohort, they adhere to the same imaging settings and protocols as the postnatal sample and will therefore allow us to comprehend how they compare to fetal subjects of the same postconceptional age, thus determining if significant differences are present between fetal and postnatal acquisitions.

Effect of scanner on fetuses. Another well-known confounding factor is the type of scanner used for MRI acquisition, which can affect the image processing algorithm and thus potentially the estimated gyrification index[108]. As previously mentioned, we used three separate cohorts to conduct our investigation: (1) our local MarsFet dataset for the first set of fetal participants, which used both 1.5 T and 3 T scanners; (2) the publicly available dHCP dataset for the second set of fetal participants, which used a 3 T Phillips scanner and; (3) the postnatal dHCP subjects, which used the same 3 T Phillips scanner as dHCP fetuses. It was not necessary to look at scanner effects in the postnatal population since only one scanner type was used. The fetal population however used three different scanners (a 1.5 T and a 3 T for MarsFet, and a 3 T for dHCP), which necessitated the fitting of three separate quadratic models: one model for the 1.5 T scanner of the MarsFet dataset, a second model for the 3 T scanner of the MarsFet dataset, and a third model for the 3 T scanner in the dHCP fetal dataset. An ANOVA was computed to statistically determine if scanner type introduced any effect. Furthermore, effect sizes and statistically significant differences were computed between all the models. The combined factors will help us determine if the trajectory of gyrification remains the same across all scanner types.

Effect of sex. Recent works, such as ref. 12 report significant effects of sex on brain growth trajectories before birth. Including sex as a categorical factor in the model is thus expected, if possible. However, since fetal sex was not systematically recorded in the clinical routine implemented at the hospital of la Timone since 2008, this information is missing in 22 participants within our MarsFet fetal dataset. We chose to proceed with a sensitivity analysis instead of excluding these 22 fetal participants. To do so, we investigated

these effects in two separate ways. (1) We included sex as a covariate in the regression discontinuity model for gyrification as a function of age, and (2) we further ran the regression discontinuity model for gyrification independently for each sex to observe any potential empirical and/or statistical differences in gyrification. These two approaches will allow us to determine if sex has an effect on the trajectory of gyrification.

Variations in image processing between BOUNTI and nnUNet segmentation approaches. To determine whether our observations could be influenced by image processing and segmentation techniques, we reproduced our entire RDD analysis using BOUNTI (Brain vOlumetry and aUtomated parcellatioN for 3D feTal MRI)[95] as a segmentation technique on our data instead of our in-house, specifically designed nnUNet model. BOUNTI is a deep-learning-based fetal brain segmentation tool introduced to accommodate the lack of robust brain segmentation methods for fetal MRI[95]. Note that this type of sensitivity analysis is particularly challenging for our results since we designed a specific segmentation approach. Therefore, in our opinion, BOUNTI is the only alternative technique to obtain accurate segmentations on our fetal dataset (MarsFet), since its effectiveness on the Fetal dHCP data was demonstrated in ref. 95. The use of BOUNTI for segmenting postnatal MRI is not recommended by the authors, yet it is the only possibility for us to ensure a consistent segmentation across our perinatal cohort. We will therefore submit our BOUNTI-segmented data to an RDD on gyrification to see if a significant discontinuity is still present regardless of variations in imaging processing.

All analyses were run using functions from the R package, including rdrobust, dplyr, and ggplot2[35,109].

## Reporting summary
Further information on research design is available in the Nature Portfolio Reporting Summary linked to this article.

## Data availability
Data for fetal and postnatal participants used in the preparation of this manuscript were obtained from the developing Human Connectome Project (dHCP), which is publicly available (https://www.developingconnectome.org/data-release/). We also used data from our local MarsFet cohort for fetal participants; however, to preserve and protect their identity, their data is not currently openly available. It can, however, be accessible upon reasonable request.

## Code availability
All relevant code is available at https://github.com/MecaLab/2025_folding_birth.git.

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

## Acknowledgements

The project leading to this publication has received funding from the Excellence Initiative of Aix-Marseille Université - A*Midex, a French "Investissements d'Avenir programme" AMX-21-IET-017. Furthermore, this project was supported by the French Agence Nationale de la Recherche (grants ANR-19-CE45-0014, ANR-21-NEU2-0005, ANR-19-CHIA-0015, ANR-22-CE45-0034) and La Fondation de France (n°00147743). Centre de Calcul Intensif d'Aix-Marseille is acknowledged for granting access to its high-performance computing resources. We would also like to thank the subjects and their families for participating in the study. The research leading to these data has also received funding from the European Research Council under the European Union Seventh Framework Programme (FP/20072013)/ERC Grant Agreement no. 319456. Additionally, the work was supported by the NIHR Biomedical Research Centres at Guys and St Thomas NHS Trust. We are grateful to the families who generously supported this trial. We would like to acknowledge that Core support for data acquisition was provided by the Wellcome/EPSRC Centre for Medical Engineering [WT 203148/Z/16/Z]. We are also thankful to the WU-Minn-Oxford Human Connectome Project consortium (1U54MH091657-01) for access to their computing resources. Data used in the preparation of this manuscript were obtained from the National Institute of Mental Health (NIMH) Data Archive (NDA). NDA is a collaborative informatics system created by the National Institutes of Health to provide a national resource to support and accelerate research in mental health. Dataset identifier(s): [Developing Human Connectome Project (dHCP) #3955]. This manuscript reflects the views of the authors and may not reflect the opinions or views of the NIH or of the Submitters submitting original data to NDA.

## Author contributions

Framing research questions, processing data, data quality check, analyzing data, and writing of the paper: A.Mih, G.A., A.P. and N.G. Data acquisition: N.G., A.Man, and M.M. Analytical and contextual contributions: G.A., A.Mih, N.G., A.P., J.L., F.R., C.D., B.D., F.B. Manuscript editing: all authors. Final approval of manuscript: all authors.

## Competing interests

The authors declare no competing interests.
