## [Transparent Peer Review file · Communications Biology]

Burst of gyrification in the human brain after birth

Corresponding Author: Dr Angeline Mihailov

Version 0:

Reviewer comments:

Reviewer #1

(Remarks to the Author)

Communications Biology

Ms. No.: COMMSBIO-24-7835-T

Title: Striking burst of gyrification in the human brain after birth

Article summary:

The authors report on the associations between postmenstrual/postconceptional age at scan and gyrification index (GI) before and shortly after birth. They do so in “819 sessions spanning 21-45 postconceptional weeks”. The authors use an analytical method accounting for discrete discontinuities in otherwise continuous associations to report on a large discontinuity in the association between age and degree of cortical folding.

Overall Impression:

The authors present on a well-motivated topic with a seemingly conceptual advance on roughly normative temporal patterns of folding in early life that are of immediate relevance to the field. While this reviewer disagrees with the strength of the authors conviction in their conclusions due to unavoidable confounding between pre- and post-natal conditions, I submit that the authors have largely performed their due diligence in attempting to address and/or model the confound out by leveraging normative fetal, preterm, and full term postnatal data and should be commended on their work. I found that the authors were mostly complete in their work (e.g., multiple segmentation approaches) and did a nice job discussion the potential mechanisms of the observed effects. Below I provide specific notes and comments that came up during my review.

Major concerns:

- Confounding design and (seemingly) overstated conclusions. There is, what seems to me, an obvious confound between the pre- and postnatal measures that is not discussed honestly at all times. At one point the authors correctly state “we cannot fully eliminate possible contamination by factors related to MRI acquisition settings” yet conclude in a heading that they observed “A striking jump of gyrification at birth that cannot be artefactual”. For example:
 - o There are differences in acquisition (e.g., dhcp fetal voxel is 260% larger than dhcp neonatal while the fetal brain also has smaller structures and likely more motion/blurring) that, while these can be mitigated by super-resolved techniques, cannot be fully removed.
 - o The neural nets used here, as I understand it, are trained on postnatal data yet are applied to fetal data (albeit with some active learning strategies). This does not necessarily mean that these segmentations are incorrect, merely that there may be some source of unobserved bias.
- “Striking effect of birth” not observed in pre-term data. The title does not apply to extremely pre-term births, and therefore implies that birth, while likely necessary, is not sufficient to elicit this effect. I do not disagree with the idea that the pre-term brain may not be primed for whatever effect birth may typically have on the brain, but it doesn't appear to me to be logically supported by the data.
- Effect of birth is unlikely to be as discrete as presented. The authors allude to “biological processes acting in the few hours/days following birth” which I think would strongly support a profound effect of birth if this can be characterized. However, as-is, the effect is modeled to be a discrete/immediate effect which is unlikely to be the case as the brain, despite being profoundly exposed to a differing environment (e.g., oxygenation, hormone surges) at birth, would not be expected to immediately fold dramatically (i.e., timescale much shorter than days as modeled). Furthermore, assuming a hydrostatic pressure driven effect (very plausible), would this be an observable process given the expected time course and available data? Especially if ~21% of folding is expected to be happening so quickly, this should be an effect that doesn't need a large effect size.
- Longitudinal data likely sufficiently powered. Similarly, the authors suggest that a longitudinal analysis in N=52 individuals

is insufficiently powered for analysis. However, given the additional power of within subject analysis and the expected effect size, this seems unlikely. Note that I see, but do not understand, the assertion in Supp Fig 14.

Minor concerns:

- The authors use the term postconceptional age throughout, but I assume the date of conception is unknown. Perhaps postmenstrual is the appropriate term?
- Is there any data on mode of delivery? Presumably, natural childbirth would impact on some of the biological processes (hormone surge, inflammation/"trauma" from delivery) potentially involved here and further support the conclusions.
- The authors conclude no sex differences (which does appear to be the case) but don't formally test for it (e.g., sex x age interaction).
- I don't see or fully understand a rationale for including the MarsFet dataset. Furthermore, its exclusion may allow for less potential confounding by site. Please justify.
- At one point there is a discussion of other factors (i.e., CSF and surface area). CSF seems relevant but GI is a function of surface area so it seems somewhat redundant to discuss.

Reviewer #2

(Remarks to the Author)

In this study, the authors present an analysis of cortical gyrification in fetal and neonatal datasets. They use the recently released dHCP neonatal and fetal cohorts combined with a smaller, local fetal MRI cohort. After training a nnUNET to segment fetal and neonatal scans, the authors compare several cortical and volumetric measures measured before and after a nominal cutoff (37 weeks gestation) demonstrating that gyrification index increases rapidly at around the time of birth. This is a quite remarkable finding and as such requires remarkable evidence to support it, however I do not find the evidence presented in the manuscript sufficiently convincing.

My main concern is that gyrification index in particular is highly sensitive to the degree of curvature in the estimated white matter surface, which in turn depends upon a series of factors such as white matter segmentation, topology corrections, image and surface smoothness etc. It is clear from the comparisons with BOUNTI that a small difference in the convexities of the WM surface can lead to a large difference in estimation of GI (Figure S12, S13). This is also evident in the discontinuous increase in WM surface area in Fig 5 and S7C. While the authors illustrate some segmentations at different ages and perform a series of sensitivity analysis, they do not do enough to show that this difference is not simply due to difference in acquisitions – 2D vs 3D - in the fetal and neonatal cohorts.

The data in Figure 6 is used to demonstrate that:

"...The congruence between preterm and fetal data supports the effectiveness of our image processing pipeline in compensating for variations in MRI acquisition. The continuity observed for preterms but not for typically developing confirms that the discontinuity we observe in gyrification is not due to differences between MRI acquisitions..."

However, I find that this data demonstrates the opposite effect: where the cortical surface is less folded at younger gestations, a putative 'acquisition effect' has no impact on GI estimation, and the in and ex utero scans overlap. As gyrification proceeds, the effect of in and ex utero scanning on GI estimation becomes apparent and at equivalent ages the ex utero scans are aligned with those acquired after 37 weeks, whereas the in utero scans are not.

It would be interesting to test the impact of using similar 2D acquisition + 3D volume reconstruction in neonatal data as a way to test the possibility that the confounding image acquisitions are behind the discrepancies in the measures.

Minor

There appears to be some confusion over the term 'post-conception'. Ages in the dHCP (fetal and neonatal) are reported in gestational weeks based on the last menstrual period, i.e.: two weeks after conception.

Reviewer #3

(Remarks to the Author)

The paper by Mihailov et al is interesting and provocative. They combine their own fetal brain MRI with dHCP data for a combined dataset of 819 sessions (both prenatal and postnatal) spanning 21 to 45 postconception weeks (PCWs). They report a very large discontinuous increase in gyrification index at birth. The paper is a nice contribution and worthy of publication pending some important revisions. There are several limitations that should be addressed and/or acknowledged. The conclusions are too strong and should be more measured. The code should be released in a clear and transparent fashion. I hope the following specific points are helpful.

- The way the results are presented currently is as if all births happen at 37 PCW. This is a big limitation of the paper in terms of 1) the data being suitable to answer the research questions and/or 2) the clarity of how the data are presented. From the plots it seems none of the datasets contain any prenatal scans from 37-41PC? First, this needs to be explained in terms of why this is a feature of the data included, or is it somehow an artifact of how the authors are choosing to display the data? Given that approximately half of births occur between 39 and 41 PCW, comparing prenatal vs postnatal brains in this time window would provide a lot of clarity about the research question.

- Since the gyrification index is a ratio, the trajectories of the component parts should be clearly displayed. From the plots presented, it seems that cortical surface area dramatically increases (which is interesting) but is it also the case (more puzzling) that the surface area of the convex hull is decreasing? This should be clarified.
- The limitation of the amount of data being excluded due to quality concerns is not adequately acknowledged. This could be a major source of bias effecting the results.
- The issue of image quality effecting automated imaging phenotypes, above and beyond making some scans unusable, is also not adequately addressed or acknowledged. It is known that variation in image quality (e.g. motion) is correlated with variation in quantitative imaging phenotypes even if all scans are "usable".
- The code release statement is inconsistent with field standards for this kind of study. The authors write that "All relevant code will be available openly upon request." But all code should be publicly released with publication. The lack seriously limits the potential impact of the work.
- The strength of the claims is just too strong given the results presented. Examples such as the following should be reconsidered: "This discovery reshapes our understanding of early brain development" "This pivotal discovery..." "visual quality assessments were conducted to eliminate all potential confounding factor" "Sensitivity analyses to rule out confounding factors". The study is interesting and raises intriguing questions. But it has numerous weaknesses and its central finding clearly needs to be replicated by an independent group.

Version 1:

Reviewer comments:

Reviewer #1

(Remarks to the Author)

The authors satisfactorily addressed the concerns raised in the review process.

Reviewer #2

(Remarks to the Author)

The authors have clarified some points of contention in their revised manuscript. Overall, the manuscript is improved. The unavoidable confounding effects of MRI acquisition across different ages and cohorts and lack of longitudinal analysis remain the major weaknesses of the study.

Additional comments are below:

1. Claims of novelty/primacy and hyperbolic language remain throughout the manuscript. At a minimum, please revise extravagant claims (e.g.: Line 36: "the neurobiological consequences of birth may hold far greater behavioral and physiological relevance than previously imagined") and hyperbolic phrasing ("strikingly", "dramatically", "far beyond" etc). While I appreciate the authors desire to project their excitement with their findings to the readership, they should consider returning to the more balanced tone used in their preprint version of this paper (<https://www.biorxiv.org/content/10.1101/2024.03.07.583908v1>), it is far more readable and presents the novel and interesting contributions of this work with clarity.

2. I do not find the argument that inconsistencies in image processing would introduce biases across all measures equally and thus absence of p-values < 0.05 across those measures indicates a lack of confounding to be compelling. There is no reason to assume that differences in specific image properties would affect estimates of e.g.: ICV in the same way it would affect estimates of white matter surface area or ventricular volume. There are multiple potential interactions along the various image processing and cortical modelling steps that may be impacted in different ways and to different extents. A lack of observed bias in subcortical volume due to MR acquisition differences does not discount an observable bias in gyrification. While unified segmentation protocols may mitigate this. There is simply no way to disentangle to confounded effects. Indeed, this unavoidable bias is acknowledged by the authors as the main limiting factor precluding a longitudinal analysis in this cohort (Line 450 - 454).

The strongly confounded effects of image acquisition and cohort/age need to be stated clearly by the authors in the limitation statement and statements that claim to otherwise account for this bias amended to acknowledge this significant experimental weakness (e.g.:

Line 223 "This ensures that the observed jump at birth is likely not due to inconsistencies in image processing, which if so, would introduce biases synonymously across all measures (tissue volumes as well as surface area and gyrification)."

Line 320: "Otherwise, if any methodological biases were responsible for the discontinuity observed in gyrification, these biases would have equally affected all remaining measures thus causing a discontinuity in all mapped features"

Line 473: We address any methodological limitations by applying unified segmentation and preprocessing techniques

across all subjects to control for MRI biases

3. The interpretations in the Discussion remain largely speculative. The argument for the role of CSF on surface area due to hydraulic pressure (Line 409) is more compelling and supported by the loss of fluid in neonates compared to fetuses. Indeed, an expansion of surface area with reduced extracerebral CSF (and thus a consequent increase in the ratio underlying GI value) proffers the most reasonable mechanism for the observed effects.

4. As noted in my previous review, and by Reviewer #3, as the area of the convex hull exhibits only minor changes over the 37w cutoff, GI is largely dependent on estimates of surface area. As a ratio measure it is critical to understand and evaluate the composite raw values of GI. To better illustrate this to the reader, plots of the composite values (SA and convex hull – currently placed in the Supplement) should be added to Figure 4 as additional panels. Likewise, in Figure 6, the composite values should be displayed to illustrate the full picture of the relationship between SA, age, hull area and GI to the readers.

Reviewer #3

(Remarks to the Author)

The response to the critiques of the original manuscript has resulted in an improved paper. Issues that remain unaddressed are related to:

-my original point 1: in explaining the lack of fetal data 37-41 weeks, the authors respond that "Acquiring cerebral MRI very close to birth poses major challenges (technical but also ethical) since the mother and fetus/newborn are particularly vulnerable during this critical period. Therefore, the sampling in time close to birth is not controlled in the present dataset, and we have very few fetuses

447 older than 37 wPC (2 in MarsFet and 4 in the dHCP)". While I agree that scanning closer to birth poses challenges, the authors overstate in the implication that such a study would essentially be impossible (to the point that this is not a major weakness of the current study design). For instance, a retrospective analysis of clinically-acquired fetal data, pooling data across institutions, could be used. The clear potential of such a study -- where prenatal and neonatal scans at the same post conception age could be directly compared -- to strengthen or disprove the hypotheses presented in the present paper should be acknowledged.

- my original point 4: The authors are correct that there is literature suggesting that subtle motion can "affect cortical feature measurements (thickness, volume, curvature) in a heterogeneous (by region) fashion." However they are incorrect in the assertion motion does not affect global measures. Motion does effect global measures. In the reference cited by the authors (Alexander-Bloch et al.) this is clearly shown in Table II. Given the potential for differential motion pre/post natal, it is a major concern that the authors seem to misunderstand this important issue.

- my original point 5: the code is not available in the github provided. the files names do not have links.

Version 2:

Reviewer comments:

Reviewer #2

(Remarks to the Author)

The authors have addressed my remaining comments.

Peer Review File:

Reviewer #1 (Remarks to the Author):

Communications Biology

Ms. No.: COMMSBIO-24-7835-T

Title: Striking burst of gyrification in the human brain after birth

Article summary:

The authors report on the associations between postmenstrual/postconceptional age at scan and gyrification index (GI) before and shortly after birth. They do so in “819 sessions spanning 21-45 postconceptional weeks”. The authors use an analytical method accounting for discrete discontinuities in otherwise continuous associations to report on a large discontinuity in the association between age and degree of cortical folding.

Overall Impression:

The authors present on a well-motivated topic with a seemingly conceptual advance on roughly normative temporal patterns of folding in early life that are of immediate relevance to the field. While this reviewer disagrees with the strength of the authors conviction in their conclusions due to unavoidable confounding between pre- and post-natal conditions, I submit that the authors have largely performed their due diligence in attempting to address and/or model the confound out by leveraging normative fetal, preterm, and full term postnatal data and should be commended on their work. I found that the authors were mostly complete in their work (e.g., multiple segmentation approaches) and did a nice job discussion the potential mechanisms of the observed effects. Below I provide specific notes and comments that came up during my review.

Major concerns:

1) Confounding design and (seemingly) overstated conclusions. There is, what seems to me, an obvious confound between the pre- and postnatal measures that is not discussed honestly at all times. At one point the authors correctly state “we cannot fully eliminate possible contamination by factors related to MRI acquisition settings” yet conclude in a heading that they observed “A striking jump of gyrification at birth that cannot be artefactual”.

We agree with the reviewer that our description of the results can assume a more subdued tone.

We do not claim we can eliminate all possible confound between pre- and post-natal measures, because some cannot be controlled (e.g. the use of different types of coil for the acquisition). Our strategy to mitigate the potential confound is by interpreting the contrast between a “continuous” trajectory for the volumetric measures, and a

42 “discontinuous” trajectory for the gyrification taking into account that all features were
43 computed based on the exact same image processing pipeline. Therefore, by stating
“A striking jump in gyrification at birth that cannot be artefactual”, we intended to
convey that idea that any biases would affect the continuous trajectories as much as
the discontinuous ones. We updated the corresponding section for the sake of
consistency and clarity, and removed the previously mentioned subtitle (page 16,
line 320).

*“Otherwise, if any methodological biases were responsible for the discontinuity*
*observed in gyrification, these biases would have equally affected all remaining measures*
*thus causing a discontinuity in all mapped features (in addition to gyrification).”*

We have also added more detail on this point in the limitations (page 21, line 435):

*“Nonetheless, this study is not without limitations. Though we do our best to accommodate*
*any present biases that may be introduced due to unavoidable differences in subject*
*environments (fetal versus postnatal) via the homogenization of processing tools, we cannot*
*entirely eliminate all effects since these populations will always be measured in different*
*environments.”*

For example:

o There are differences in acquisition (e.g., dhcp fetal voxel is 260% larger than dhcp
neonatal while the fetal brain also has smaller structures and likely more motion/blurring)
that, while these can be mitigated by super-resolved techniques, cannot be fully removed.

Indeed this is an important point, yet, we still see continuity in most features, and we
compare the effect size across features computed from the exact same image
processing pipeline. This constitutes in our opinion a pragmatic strategy to
compensate for all uncontrolled(-able) factors related to image quality and
processing, including variations in voxel size.

o The neural nets used here, as I understand it, are trained on postnatal data yet are
applied to fetal data (albeit with some active learning strategies). This does not necessarily
mean that these segmentations are incorrect, merely that there may be some source of
unobserved bias.

Indeed, however in the present study this was the best method we could apply and
do not claim that all potential biases were removed. And again, we are interested in
the contrasting continuous and non-continuous results, therefore any unobserved
bias that would be present should have equally affected all features.

**2)** “Striking effect of birth” not observed in pre-term data. The title does not apply to
extremely pre-term births, and therefore implies that birth, while likely necessary, is not

sufficient to elicit this effect. I do not disagree with the idea that the pre-term brain may not
be primed primed for whatever effect birth may typically have on the brain, but it doesn't
appear to me to be logically supported by the data.

We agree that our claim of "striking effect of birth" do not apply directly to pre-term
data, and in particular to extremely preterm births due to the presence of less folded
brains at earlier fetal ages. Please see our answers to concern #2 from Reviewer 2
(page 9) and #2 from Reviewer 3 below (page 14). We provide new RDD analyses
showing that the effect of birth is also very high at 36 and 35 wPC, and new data
with the measure of the area of the convex hull.

Regarding preterms at younger age, we agree that the gap between postnatal and
prenatal data becomes smaller, which supports an analysis focusing on relative
values instead of absolute values (size of the gap relative to the increase in GI since
20wPC for instance).

More importantly, as pointed out by the reviewers, such an analysis should consider
potential interactions between birth and brain folding *dynamics*, since brain folding
accelerates in particular between 20 and 30wPC. This is visible on the new figure
showing the area of the convex hull, showing that the area of the white surface and
the area of the convex hull remain close between 20 and 25wPC, and diverge
between 25 and 30wPC. In our opinion, entering into such considerations would
extend the work far beyond the scope of this paper and therefore we do not dive
deeper into this discussion.

Overall, the current study focuses on the "normal" trajectory, though we agree that
further and more refined work is required to better understand the interaction
between birth and gyrification in very preterm brains as well. To be more precise, we
specified in the manuscript that the jump in gyrification at birth was observed at-term
birth in our study (page 10, line 201; page 15, line 309): added "*at-term*".

**3)** Effect of birth is unlikely to be as discrete as presented. The authors allude to "biological
processes acting in the few hours/days following birth" which I think would strongly support
a profound effect of birth if this can be characterized. However, as-is, the effect is modeled
to be a discrete/immediate effect which is unlikely to be the case as the brain, despite being
profoundly exposed to a differing environment (e.g., oxygenation, hormone surges) at birth,
would not be expected to immediately fold dramatically (i.e., timescale much shorter than
120 days as modeled). Furthermore, assuming a hydrostatic pressure driven effect (very
plausible), would this be an observable process given the expected time course and
available data? Especially if ~21% of folding is expected to be happening so quickly, *this*
*should be an effect that doesn't need a large effect size.*

We agree that the effect of birth is unlikely to be discrete, as with any biological
process acting at the 'organ level'. Our aim is to statistically model a sharp transition,
which is often challenging in biological systems. In the present work, we chose to
use an RDD model that assesses potential discontinuity in the measures because it
is well fitted to the sampling in time that is imposed by the data. Indeed, the use of
other methodological approaches for assessing sharp changes in a given trajectory
such as those based on second or third order features (speed and acceleration)
require a high and regular sampling in time, in particular around the timing of the
event of interest, which is birth in our case. Since the timing of fetal and postnatal
acquisitions were not controlled with the aim of having high sampling close to birth in
the present dataset, we cannot accurately estimate speed and acceleration close to
birth. Note that acquiring cerebral MRI very close to birth poses major challenges
(technical but also ethical) since the mother and fetus/newborn are particularly
vulnerable during this critical period. Therefore, the uncontrolled sampling in time in
the present dataset imposes the use of statistical modeling based on discontinuity.
Future work will be required to closely investigate the continuity vs. discontinuity
aspects of the sharp transition we report, but given that specifically acquired data is
necessary for that, we consider such an investigation as beyond the scope of
present study.

To conclude, we also agree on the observation from Reviewer 1 that if we had
access to an appropriate sampling in time, we would observe a sharp but
progressive change, thus a period along the trajectory may not show a very large
effect size at a single time point but instead a *relatively high effect size during*
*several time points in a row*. We added more text in the manuscript to describe our
intentions when using the term "discontinuity" (**page 18, line 380**):

*"Note that in the present work, we chose to use an RDD model that assesses potential*
*discontinuity in the measures because it is well fitted to the sampling in time that is imposed*
*by the data. Since the timing of fetal and postnatal acquisitions were not controlled with the*
*aim of having high sampling close to birth in the present dataset, we cannot accurately*
*estimate speed and acceleration close to birth. Future work will be required to more closely*
*investigate the continuity versus discontinuity aspects of the sharp transition we report in*
*the present study"*

**4)** Longitudinal data likely sufficiently powered. Similarly, the authors suggest that a
longitudinal analysis in N=52 individuals is insufficiently powered for analysis. However,
given the additional power of within subject analysis and the expected effect size, this
seems unlikely. Note that I see, but do not understand, the assertion in Supp Fig 14.

Though n = 52 does seem like a reasonable amount of subjects for a longitudinal
study in general, it would not be sufficient for the current study due to the quantity of
data points required to assess our effect. In order to be able to capture any effect

that may occur at birth, we would need fine-grained data (several data points) in the
167 weeks just before and just after birth, which would be near impossible in the present
168 days. Most of the longitudinal timepoints we have are several weeks, even months
apart which could never provide us with any information on what is happening
specifically around the time of birth. Overall, the limitation is not on the n subjects
that have longitudinal data, but rather on the number of timepoints per subject. At the
very minimum, to estimate even the first derivative of a trajectory, we would need at
least 3+ timepoints. In the present dataset, only 5 subjects have 3 timepoints, which
is unfortunately insufficient to make a quantitative conclusion about anything.

The purpose of supplementary figure 15 is to display the connection (lines) between
the time points per subject (data points), with each colour referring to a specific time
point number. Specifically, all gray points have 1 time point, therefore no longitudinal
data, all the blue points represent the first time point of those with longitudinal data,
all yellow points represent the second time point of those with longitudinal data, and
so on. We added additional text to the manuscript in hopes of clarifying this
explanation (supplementary page 17, line 261; supplementary page 13, line
161).

*“Supplementary Figure 15 illustrates the number of timepoints and their age along the trajectory*
*of subjects with longitudinal data available in our cohort.”*

***Added to Supplementary Figure 15 caption:***

*“Specifically, blue points represent the first timepoint in subjects with longitudinal data,*
*yellow points represent the second timepoint, orange points represent the third timepoints*
*and pink points represent the fourth timepoints. As shown, not many subjects have more*
*than two timepoints, and even less have timepoints before and after birth.”*

Minor concerns:

**1)** The authors use the term postconceptional age throughout, but I assume the date of
conception is unknown. Perhaps postmenstrual is the appropriate term?

In the MarsFet cohort, we estimated the conception date using ultrasound, which is
supposed to be the most accurate form of estimation, even more so than
postmenstrual age. Therefore, along with the dHCP, which does use postmenstrual
age, we decided to adopt the term “postconceptional” throughout this study.

Additional text was added to the manuscript to help explain this in greater detail
(page 6, line 123).

*“To note, the date of conception is estimated using ultrasound in the MarsFet cohort, and the*
*postmenstrual date (i.e., 2 weeks after the last menstrual period) in the fetal dHCP cohort. In*
*this study however, ages are collectively referred to as weeks post conception (wPC).”*

**2)** Is there any data on mode of delivery? Presumably, natural childbirth would impact on
some of the biological processes (hormone surge, inflammation/"trauma" from delivery)
potentially involved here and further support the conclusions.

This is a very important question and therefore a factor to consider that would
provide valuable information in a future study. Some information related to delivery
are available for the dHCP data, and we are working to get this information for
MarsFet, but the data is incomplete at this stage.

**3)** The authors conclude no sex differences (which does appear to be the case) but don't
formally test for it (e.g., sex x age interaction).

We did test for sex differences by submitting subjects with sex data to an RDD
analysis that included sex as a factor. Furthermore, we also ran individual RDDs to
confirm a large effect size for gyrification at birth in both males and females. Both by
including sex as a factor in the model and by running an independent RDD for each
sex confirms that there were no influences of sex on the sharp change of gyrification
at birth.

Nonetheless, we assessed for potential interactions in the fetal and postnatal
populations, independently. As show on the figures below, we observe no significant
interactions between age*sex in neither fetal nor postnatal neurodevelopment during
the perinatal time frame:

*Fetal population model output for interaction term is p-val = 0.5591*

Postnatal population model output for interaction term is $p\text{-val} = 0.1988$

Entire perinatal population model output for interaction term is $p\text{-val} = 0.7937$

**4)** I don't see or fully understand a rationale for including the MarsFet dataset. Furthermore,
its exclusion may allow for less potential confounding by site. Please justify.

The addition of the MarsFet cohort in this study is instrumental since it significantly
increases our sample size, which is essential in improving the balance between fetal
and postnatal data in the RDD models. In addition, it allows us to assess the
influence of acquisition settings on the features extracted from fetuses in order to
confirm that the effect size is much lower than the effect on birth (which is our main
result). Furthermore, the variations in acquisition settings between the MarsFet and
dHCP include critical changes in the way the MRI is done since MarsFet scans are
acquired in clinical settings, while the dHCP scans are acquired in research-oriented

settings. Thus, what is labeled as “variations in acquisition settings” goes far beyond
simple variations in MRI sequences. Therefore, the minimal variation seen when
testing for potential differences between MarsFet and dHCP in fetuses is a further
confirmation of the robustness of our image processing pipeline and our results. This
information was added to the manuscript (page 25, line 518).

*“The addition of the MarsFet cohort in this study is instrumental since it significantly
increases our sample size, which is essential in improving the balance between fetal and
postnatal data in the RDD models. It also allows us to assess the influence of acquisition
settings on the features extracted from fetuses since the variations in acquisition settings
between the MarsFet and dHCP include critical changes in the way the MRI is acquired.
Specifically, MarsFet scans are acquired in clinical settings, while the dHCP scans are
acquired in research-oriented settings. Thus, what is labeled as “variations in acquisition
settings” goes far beyond simple variations in MRI sequences.”*

**5)** At one point there is a discussion of other factors (i.e., CSF and surface area). CSF
seems relevant but GI is a function of surface area so it seems somewhat redundant to
discuss.

We understand the point of the Reviewer here, but do not fully agree that surface area is
redundant since it is a component of gyrification. Furthermore, understanding the
dynamics of surface area relative to the other component, the convex hull, can aid in the
explanation of our results as will be discussed in more detail below (answer #2 to
Reviewer 2 (page 9); and answer #2 to Reviewer 3 (page 14)).

**Reviewer #2 (Remarks to the Author):**

In this study, the authors present an analysis of cortical gyrification in fetal and neonatal
datasets. They use the recently released dHCP neonatal and fetal cohorts combined with a
smaller, local fetal MRI cohort. After training a nnUNET to segment fetal and neonatal
scans, the authors compare several cortical and volumetric measures measured before and
after a nominal cutoff (37 weeks gestation) demonstrating that gyrification index increases
rapidly at around the time of birth. This is a quite remarkable finding and as such requires
remarkable evidence to support it, however I do not find the evidence presented in the
manuscript sufficiently convincing.

**1)** My main concern is that gyrification index in particular is highly sensitive to the degree of
curvature in the estimated white matter surface, which in turn depends upon a series of
factors such as white matter segmentation, topology corrections, image and surface
smoothness etc. It is clear from the comparisons with BOUNTI that a small difference in the
convexities of the WM surface can lead to a large difference in estimation of GI (Figure S12,
S13). This is also evident in the discontinuous increase in WM surface area in Fig 5 and

S7C. While the authors illustrate some segmentations at different ages and perform a series
of sensitivity analysis, they do not do enough to show that this difference is not simply due
to difference in acquisitions – 2D vs 3D - in the fetal and neonatal cohorts.

In the present work, we purposely use the global gyrification index that is computed
as the ratio between *total* white surface area and *total* area of the convex hull. This
global feature is much more robust than local gyrification index as seen in¹.

In addition, reproducing our main result (jump of GI at birth) with a completely
different segmentation technique (nnUnet vs BOUNTI) is a strong demonstration of
the limited impact of different image processing aspects involved in the full pipeline.
With both segmentation techniques, we see continuity in most features computed,
which constitute in our opinion an empirical demonstration indicating that any
potential differences due to acquisitions were compensated for. We are interested in
the contrasting continuous and non-continuous results, therefore any unobserved
bias that would be present should have equally affected all features. With the 2
pipelines we observe a large effect size for the GI but a continuous (therefore non-
significant RDD outcome) for other features such as the cortical gray matter.

The reviewer referred to figure S7C, which is the CSF. The CSF indeed shows a
larger discontinuity at birth. Regarding the surface area, we agree that the
discontinuity can be debated. However, our results (figure 5, S7A) show that the
effect size for gyrification is considerably larger than for surface area in all analyses
(and across segmentation methods). This can also lead us to interpret the trajectory
for surface area at birth as more of an inflection than a significant jump. The
trajectory of surface area therefore does warrant a further investigation in a separate
study, once datasets with higher sampling in time will be available (see also our
response to Reviewer #1 concern 3).

**2)** The data in Figure 6 is used to demonstrate that:

"...The congruence between preterm and fetal data supports the effectiveness of our image
processing pipeline in compensating for variations in MRI acquisition. The continuity
observed for preterms but not for typically developing confirms that the discontinuity we
observe in gyrification is not due to differences between MRI acquisitions..."

However, I find that this data demonstrates the opposite effect: where the cortical surface is
less folded at younger gestations, a putative 'acquisition effect' has no impact on GI
estimation, and the in and ex utero scans overlap. As gyrification proceeds, the effect of in
and ex utero scanning on GI estimation becomes apparent and at equivalent ages the ex
utero scans are aligned with those acquired after 37 weeks, whereas the in utero scans are
not.

We agree that our description and interpretation of Figure 6 can be improved in
order to avoid confusion.

If we correctly understand the concern from the reviewer, the interpretation of Figure
6 would be that the use of different MRI acquisitions would impact differently young,
smoother brains than older, folded brains. We disagree with this interpretation
because:

- 1) In both fetuses and preterms, the evolution of GI with age is smooth and
continuous, in agreement with the marked changes in brain folding observed
in the anatomical scans. So we observe that GI evolves as expected with age
and magnitude of folding, within each scanner, i.e. for each acquisition
setting. This means that when a brain is smooth, GI is lower than when a
brain is folded, for each acquisition setting used in this study.
- 2) We explicitly tested for acquisition effect across the fetus dataset (Fig. S8 and
corresponding section 2.5.2), which revealed very low effect sizes across all
scanner comparisons (*MarsFet_1.5T vs MarsFet_3T = 0.24*, *MarsFet_1.5T vs*
*dHCP_3T = 0.09*, *MarsFet_3T vs dHCP_3T = 0.15*). There was a minor
significant effect when comparing 1.5T vs 3T scanners between the cohorts,
but no significant difference between the 3T scanners of the MarsFet and
dHCP cohorts. Note that the variations in acquisition settings between the
fetuses from MarsFet and the fetuses from dHCP include critical variations in
the way the MRI is done since MarsFet scans are acquired in clinical settings,
while the dHCP scans are acquired in research-oriented settings. Thus, what
is labeled as “variations in acquisition settings” goes far beyond simple
variations in MRI sequences.
- 3) The size of the effect of birth on GI is massive compared to the size of the
effect of the scanner across the fetus dataset (previous point).
- 4) The effect of birth on the brain is not only visible on the features we report in
our analyzes but also clearly visible from the reconstructed 3D anatomical
volumes, and in particular the strong reduction in CSF (refer to the figure
below).
- 5) We conducted extensive visual assessments of the reconstructed 3D
volumes, of the segmentation and of the white surface, blind to the age and
thus to the type of sequence, and did not identify any putative interaction
between the magnitude of folding and image or segmentation quality.
- 6) The continuous trajectory for most volume measures (especially the cortical
gray matter and the white matter volumes) across different ages for fetuses
and neonates support the absence of an interaction between age and a
putative ‘acquisition effect’.

We ask the reviewer to refer to a paper by Levéfre et al., 2016 where they show
direct comparisons of subjects’ anatomical images from the different populations
(fetuses and preterms) at equivalent postconceptional ages, where it is visible that
preterm subjects show a more folded brain².

**3)** It would be interesting to test the impact of using similar 2D acquisition + 3D volume
reconstruction in neonatal data as a way to test the possibility that the confounding image
acquisitions are behind the discrepancies in the measures.

We agree on the principle, but note that testing for the impact of using 2D and 3D
volume acquisitions would unfortunately not be sufficient since other important
factors are also different (voxel size and slice thickness in the acquired 2D
acquisitions, but also the type of coil used for the acquisition) thus using the same
2D/3D reconstruction technique would not fully compensate for all these factors. We
previously tried to apply NESVOR to the dHCP data and got lower quality
reconstructions. This is not surprising since the reconstruction method was designed
and validated on the dHCP dataset³. Therefore, such analyses would serve as an
assessment of the impact of 2D/3D reconstructions in general, which is not the
scope of the present study. Furthermore, we conducted extensive visual
assessments leading us to keep only high quality data.

As for concern 1), we see continuity in most features computed, which constitute in
our opinion an empirical demonstration indicating that these differences due to
acquisitions were compensated for. By comparing the impact of birth on different
features extracted from the same pipeline, we account for these confounding factors.

Minor

**1)** There appears to be some confusion over the term 'post-conception'. Ages in the dHCP
(fetal and neonatal) are reported in gestational weeks based on the last menstrual period,
i.e.: two weeks after conception.

The meaning of the term "postconceptional" throughout the paper has been raised
by other reviewers as well. This was clarified in the manuscript (**page 6, line 123**).
Briefly, the dHCP estimated the gestational age as 2 weeks *after* the last menstrual
period, while the MarsFet estimated the age of their fetuses based on ultrasound.
We collectively refer to this as postconceptional age.

*"To note, the date of conception is estimated using ultrasound in the MarsFet cohort, and the*
*postmenstrual date (i.e., 2 weeks after the last menstrual period) in the fetal dHCP cohort. In*
*this study however, ages are collectively referred to as weeks post conception (wPC)."*

**Reviewer #3 (Remarks to the Author):**

The paper by Mihailov et al is interesting and provocative. They combine their own fetal
brain MRI with dHCP data for a combined dataset of 819 sessions (both prenatal and
postnatal) spanning 21 to 45 postconception weeks (PCWs). They report a very large
discontinuous increase in gyrification index at birth. The paper is a nice contribution and

worthy of publication pending some important revisions. There are several limitations that
should be addressed and/or acknowledged. The conclusions are too strong and should be
more measured. The code should be released in a clear and transparent fashion. I hope the
following specific points are helpful.

**1)** The way the results are presented currently is as if all births happen at 37 PCW. This is a
big limitation of the paper in terms of 1) the data being suitable to answer the research
questions and/or 2) the clarity of how the data are presented. From the plots it seems none
of the datasets contain any prenatal scans from 37-41PC? First, this needs to be explained
in terms of why this is a feature of the data included, or is it somehow an artifact of how the
authors are choosing to display the data? Given that approximately half of births occur
between 39 and 41 PCW, comparing prenatal vs postnatal brains in this time window would
provide a lot of clarity about the research question.

We thank the Reviewer for raising this excellent point. We provide our answer in two
parts: A) data-related aspects and B) statistical modeling aspects:

439 A) data-related aspects:

37 wPC was chosen based the WHO reference stating 37 as the minimal
postconceptional age before being considered preterm⁴.

The absence of prenatal scans from 37-41 wPC is due to two factors:

1) Acquiring cerebral MRI very close to birth poses major challenges (technical
but also ethical) since the mother and fetus/newborn are particularly
vulnerable during this critical period. Therefore, the sampling in time close to
birth is not controlled in the present dataset, and we have very few fetuses
older than 37 wPC (2 in MarsFet and 4 in the dHCP).

2) In order to minimize potential bias related to image processing, we excluded
all the subjects for which the quality of either the reconstructed volume, tissue
segmentation and white surface was not optimal. The 6 fetuses aged 37+
wPC did not pass the quality control. A concern could then be a potential bias
related to a lower rate of success in our image processing pipeline, but this
would affect only 6 subjects out of several hundreds, which is unlikely, and
impossible to assess properly in absence of more data acquired close to
birth. We therefore focused on statistical modeling aspects as detailed below.

B) statistical modeling aspects

We did implement a series of analyses in order to assess the potential influence of
the cut-off value in RDD models. We reproduced the RDD analyzes with a cut-off set
at 35 or 36 wPC. For the analysis with a cut-off at 36 wPC, we excluded fetal
subjects older than 36 postconceptional weeks and preterm subjects younger than
36 wPC. This allowed us to simulate a cohort with a birth cut-off of 36 wPC that has
only fetal subjects before 36 wPC, and only postnatal (albeit including moderate
preterms) after 36 wPC. And we reproduced the analysis with a cut-off set at 35

wPC. As illustrated in the figures below, we still observed a significant jump at birth
with a high effect size in both analyzes. Extending these sensitivity analyses to later
postconceptional ages (38, 39, etc.) would be optimal, but it is impossible with the
data available today.

Effect size = 14.19 (p value = 9.74×10^{-46})

Effect size = 10.09 (p value = 7.78×10^{-24})

Note that extending the analysis to younger age (e.g. cut-off at 30 wPC) would be instrumental in clarifying the impact of birth on the preterm brain, but the interpretation would become very challenging because of the potential interaction between the very active dynamics of folding at these developmental stages and the effect of birth. This remains to be investigated in future work.

In conclusion, since this point is very important, we decided to include an explanation on this in the limitations section (**page 21, line 439**).

*“Additionally, we focused on a birth age cut-off of 37 weeks post-conception (wPC) since this is*
*the threshold for being considered at-term based on references from the World Health*

*Organization*². Moreover, this study did not include fetal participants older than 37 weeks due to
several challenges. These challenges include limited access, primarily due to ethical
considerations, as well as the exclusion of the few available older participant scans due to lower
image quality. Despite these limitations, setting the cut-off at 37 wPC still enabled us to confirm
the impact of birth on brain anatomy in a typically developing population. As more data become
available, it would be valuable to extend the analysis to include older at-term age cut-offs and
potentially even non-term age cut-offs using RDD analyses.”

**2)** Since the gyrification index is a ratio, the trajectories of the component parts should be
clearly displayed. From the plots presented, it seems that cortical surface area dramatically
increases (which is interesting) but is it also the case (more puzzling) that the surface area
of the convex hull is decreasing? This should be clarified.

This is a very important point that indeed was missing in the original submission. We
now provide the additional data and figure presented below, showing the
independent trajectories of the GI components (surface area and convex hull, both
nnUnet and BOUNTI segmentations). RDD shows that the impact of birth on the
convex hull is non-significant (*effect size = 1.3, p-value = 0.2*). This new data shows
the sharp increase in GI is a consequence of an acceleration in the expansion of the
surface area while the expansion of the convex hull is rather limited.

nnUnet segmentation:

BOUNTI segmentation:

We included this new analysis and text in the manuscript (page 12, line 231; page 18, line 369; Supplementary page 7 (Supplementary Figure 8), line 82).

“Gyrification is a measure composed of the ratio between two components: the surface area and the convex hull. Since the surface area is already discussed above, we decided to also run an RDD on the convex hull in order to help us explain the jump that we are observing in gyrification at birth. The convex hull did not show a high effect size for a discontinuity at birth (effect size = 1.26, $p = 0.21$)(Supplementary Figure 8).”

“Furthermore, since the surface area component of gyrification accelerates faster than the convex hull in growth as it approaches birth, biomechanical factors may also be at play revealing underlying dynamics of gyrification. Compared to the convex hull, faster surface area acceleration likely plays a bigger role in constraining the brain into a restricted space.”

Also to expand on the mechanical influence that birth can have on the brain, we added the following text which discusses the rapid expansion of early gyrification and references a paper showing how during birth, molding temporarily compresses certain areas of the skull via intense pressure from passing through the mother’s birth canal⁵ (page 18, line 373).

*"During birth, molding temporarily compresses certain areas of the skull via intense*
*pressure from passing through the mother's birth canal. Within a few weeks, the brain*
*rounds out in healthy subjects. At the same time, the early neonatal period is a dynamic*
*period for rapid gyrification expansion and development to ensure normal growth and*
*therefore normal cognitive and motor development. The event of molding, in addition to*
*several other biological and environmental factors could help contribute to a plausible*
*interpretation of why we see changes in gyrification after birth."*

**3)** The limitation of the amount of data being excluded due to quality concerns is not
adequately acknowledged. This could be a major source of bias effecting the results.

Compared to the amount of data excluded in postnatal imaging, there are usually
more scan exclusions of fetal participants due to uncontrolled motion in the womb,
which is not surprising. We tried to the best of our ability to include only great quality
fetal scans without artefacts in order to limit as much source of bias as we can. We
added this information to the manuscript (page 5, line 110). There are several
papers aimed at developing methods to improve data quality in fetal acquisitions⁵,
however this topic is beyond the scope of our paper.

*"Fetal scans typically result in greater exclusion numbers due to uncontrolled motion in the*
*womb, which is not surprising. To the best of our ability, we included only good quality fetal*
*scans without artefacts in order to limit as much source of bias as possible."*

**4)** The issue of image quality effecting automated imaging phenotypes, above and beyond
making some scans unusable, is also not adequately addressed or acknowledged. It is
known that variation in image quality (e.g. motion) is correlated with variation in quantitative
imaging phenotypes even if all scans are "usable".

This is an accurate point brought up by the reviewer. There is literature that
discusses this concept, stating that subtle motion can affect cortical feature
measurements (thickness, volume, curvature) in a heterogeneous (by region)
fashion⁶. While that is an important factor, we investigate global features. This would
however definitely be necessary to control for in cortical studies investigating
parcellated regions.

In the present work, however, our point is that if such a bias were present in "usable"
scans, it would affect all features simultaneously (both the gyrification and those that
are continuous), and changes at the global level would be minimal. We assess the
impact of birth by comparing the effect size across different features extracted from
the same pipeline. This constitutes, in our opinion, a pragmatic strategy to
compensate for all uncontrolled(-able) factors related to image quality and
processing.

5) The code release statement is inconsistent with field standards for this kind of study. The authors write that "All relevant code will be available openly upon request." But all code should be publicly released with publication. The lack seriously limits the potential impact of the work.

We agree with this point and will release the code in conjunction with the resubmission (https://github.com/MecaLab/2025_folding_birth.git).

6) The strength of the claims is just too strong given the results presented. Examples such as the following should be reconsidered: "This discovery reshapes our understanding of early brain development" "This pivotal discovery..." "visual quality assessments were conducted to eliminate all potential confounding factor" "Sensitivity analyses to rule out confounding factors". The study is interesting and raises intriguing questions. But it has numerous weaknesses and its central finding clearly needs to be replicated by an independent group.

We agree and will tone down our assertions in the revised manuscript (**for example page 2, line 36; page 4, line 87; page 4, line 90; page 12, line 254**). However, we still maintain that when these results will be replicated and verified with animal studies, this work will be considered pivotal in the field since it will be the first to quantitatively characterize the impact of birth on brain anatomy.

References

1. Schaer, M. *et al.* A Surface-Based Approach to Quantify Local Cortical Gyrification. *IEEE*
*Trans. Med. Imaging* **27**, 161–170 (2008).

2. Lefèvre, J. *et al.* Are Developmental Trajectories of Cortical Folding Comparable Between
Cross-sectional Datasets of Fetuses and Preterm Newborns? *Cereb. Cortex* **26**, 3023–3035
(2016).

3. Cordero-Grande, L. *et al.* Sensitivity Encoding for Aligned Multishot Magnetic Resonance
Reconstruction. *IEEE Trans. Comput. Imaging* **2**, 266–280 (2016).

4. World Health Organization. *The WHO Application of ICD-10 to Deaths during the Perinatal*
*Period: ICD-PM.* (World Health Organization, Geneva, 2016).

5. Uus, A. *et al.* Scanner-based real-time three-dimensional brain + body slice-to-volume
reconstruction for T2-weighted 0.55-T low-field fetal magnetic resonance imaging. *Pediatr.*
*Radiol.* (2025) doi:10.1007/s00247-025-06165-x.

6. Alexander-Bloch, A. *et al.* Subtle in-scanner motion biases automated measurement of brain
anatomy from in vivo MRI. *Hum. Brain Mapp.* **37**, 2385–2397 (2016).

Peer Review File:

Reviewer #1 (Remarks to the Author):

The authors satisfactorily addressed the concerns raised in the review process.

Reviewer #2 (Remarks to the Author):

The authors have clarified some points of contention in their revised manuscript. Overall, the manuscript is improved. The unavoidable confounding effects of MRI acquisition across different ages and cohorts and lack of longitudinal analysis remain the major weaknesses of the study.

Additional comments are below:

1. Claims of novelty/primacy and hyperbolic language remain throughout the manuscript. At a minimum, please revise extravagant claims (e.g.: Line 36: “the neurobiological consequences of birth may hold far greater behavioral and physiological relevance than previously imagined”) and hyperbolic phrasing (“strikingly”, “dramatically”, “far beyond” etc). While I appreciate the authors desire to project their excitement with their findings to the readership, they should consider returning to the more balanced tone used in their preprint version of this paper (<https://www.biorxiv.org/content/10.1101/2024.03.07.583908v1>), it is far more readable and presents the novel and interesting contributions of this work with clarity.

In the revised manuscript, we further toned down our claims and extended the description of the major weakness within the study. We tried to find the right balance between sharing our excitement and limiting hyperbolic language by making a few more changes (page 2, line 36; page 26, line 539; page 9, line 186; page 4, line 88, page 15, line 308):

“This finding sheds light onto the understanding of early brain development, suggesting that the neurobiological consequences of birth may hold significant behavioral and physiological relevance.”

“Thus, what is labeled as “variations in acquisition settings” goes beyond simple variations in MRI sequences.”

“To empirically and quantitatively comprehend this noticeable shift in the trajectory...”

“Our findings provide new insights into the biological impact of birth on the newborn brain, highlighting influences on gyrification.”

“...we report a substantial jump of 21.4% in gyrification around the time of at-term birth.”

We still, however, maintain our claim regarding the novelty of our study in reporting the strong impact of birth on brain folding.

2. I do not find the argument that inconsistencies in image processing would introduce biases across all measures equally and thus absence of p-values < 0.05 across those measures indicates a lack of confounding to be compelling. There is no reason to

assume that differences in specific image properties would affect estimates of e.g.: ICV in the same way it would affect estimates of white matter surface area or ventricular volume. There are multiple potential interactions along the various image processing and cortical modelling steps that may be impacted in different ways and to different extents. A lack of observed bias in subcortical volume due to MR acquisition differences does not discount an observable bias in gyrification. While unified segmentation protocols may mitigate this. There is simply no way to disentangle to confounded effects. Indeed, this unavoidable bias is acknowledged by the authors as the main limiting factor precluding a longitudinal analysis in this cohort (Line 450 - 454).

The strongly confounded effects of image acquisition and cohort/age need to be stated clearly by the authors in the limitation statement and statements that claim to otherwise account for this bias amended to acknowledge this significant experimental weakness (e.g.:

We agree that there is simply no way to disentangle confounded effects in the present study.

Our point is not that the bias is fully accounted for, but that it is very unlikely that it could affect the surface area and convex hull measures much more than the others, which would be required to generate such a large effect for GI compared to *all the other measures* (effect sizes reported on Fig 5).

We have changed some of the wording in our manuscript in order to clarify our stance on this point (changes below).

Change in manuscript (**page 21, line 436**):

“Specifically, the proposed unified image processing approach can mitigate but not fully compensate for the confounding effects of image acquisition, cohort or age. Indeed, there is simply no way to disentangle confounding effects related to in-utero versus ex-utero MRI acquisition.”

Line 223 “This ensures that the observed jump at birth is likely not due to inconsistencies in image processing, which if so, would introduce biases synonymously across all measures (tissue volumes as well as surface area and gyrification).”

Change in manuscript (**page 11, line 229**):

“This supports the assumption that the observed jump at birth is not due to inconsistencies in image processing. It is unlikely that such a bias would specifically target surface area and convex hull measures and thus induce a much higher effect size compared to all other measures submitted to the same image processing pipeline”

Line 320: “Otherwise, if any methodological biases were responsible for the discontinuity observed in gyrification, these biases would have equally affected all remaining measures thus causing a discontinuity in all mapped features”

Change in manuscript (**page 16, line 319**):

“Otherwise, if methodological biases were responsible for the discontinuity observed in gyrification, these biases would likely have introduced some form of discontinuity in all mapped features (in addition to gyrification).”

Line 473: We address any methodological limitations by applying unified segmentation and preprocessing techniques across all subjects to control for MRI biases

Change in manuscript (page 22, line 476):

"We proposed a unified segmentation and preprocessing technique across all subjects to help mitigate inevitable biases related to MRI acquisition."

3. The interpretations in the Discussion remain largely speculative. The argument for the role of CSF on surface area due to hydraulic pressure (Line 409) is more compelling and supported by the loss of fluid in neonates compared to fetuses. Indeed, an expansion of surface area with reduced extracerebral CSF (and thus a consequent increase in the ratio underlying GI value) proffers the most reasonable mechanism for the observed effects.

We agree, our aim is to encourage meaningful discussions throughout the community by drawing attention to the many mechanisms related to birth that may have a greater than expected impact on the brain, and that remain under-explored.

4. As noted in my previous review, and by Reviewer #3, as the area of the convex hull exhibits only minor changes over the 37w cutoff, GI is largely dependent on estimates of surface area. As a ratio measure it is critical to understand and evaluate the composite raw values of GI. To better illustrate this to the reader, plots of the composite values (SA and convex hull – currently placed in the Supplement) should be added to Figure 4 as additional panels. Likewise, in Figure 6, the composite values should be displayed to illustrate the full picture of the relationship between SA, age, hull area and GI to the readers.

We agree, we added an additional panel to figure 4 to the main text that illustrates the components of gyrification (see page 10, line 203-204).

On the other hand, we prefer to not add the components onto Figure 6 since we do not want to distract readers from the purpose of that figure, which is to show the neurodevelopment of gyrification in preterms alongside that of typical subjects.

Reviewer #3 (Remarks to the Author):

The response to the critiques of the original manuscript has resulted in an improved paper. Issues that remain unaddressed are related to:

-my original point 1: in explaining the lack of fetal data 37-41 weeks, the authors respond that "Acquiring cerebral MRI very close to birth poses major challenges (technical but also ethical) since the mother and fetus/newborn are particularly vulnerable during this critical period. Therefore, the sampling in time close to birth is not controlled in the present dataset, and we have very few fetuses 447 older than 37 wPC (2 in MarsFet and 4 in the dHCP)". While I agree that scanning closer to birth poses challenges, the authors overstate in the implication that such a study would essentially be impossible (to the point that this is not a major weakness of the current study design). For instance, a retrospective analysis of clinically-acquired fetal data, pooling data across institutions, could be used. The clear potential of such a study -- where prenatal and neonatal scans at the same post conception age could be

directly compared -- to strengthen or disprove the hypotheses presented in the present paper should be acknowledged.

We attempted to do exactly as the reviewers suggested, i.e., retrospectively pooling such data across centers accessible to us, however, as stated in our manuscript, we could not acquire a large amount of fetal subjects older than 37 wPC.

We agree with the reviewer's point and added a line explaining the importance of acquiring sufficient data representing the weeks closest to birth so that neonatal and fetal scans of the same wPC can be directly compared (**see page 21, line 447**).

"Future studies involving sufficient data points representing the weeks closest to birth to allow for a direct comparison between fetal and postnatal subjects at the same postconceptional ages is necessary to strengthen or disprove the hypotheses presented in the present paper."

- my original point 4: The authors are correct that there is literature suggesting that subtle motion can "affect cortical feature measurements (thickness, volume, curvature) in a heterogeneous (by region) fashion." However they are incorrect in the assertion motion does not affect global measures. Motion does effect global measures. In the reference cited by the authors (Alexander-Bloch et al.) this is clearly shown in Table II. Given the potential for differential motion pre/post natal, it is a major concern that the authors seem to misunderstand this important issue.

We understand the importance of this issue and do not underestimate the potential influence of subtle motion on our results.

Our points are:

1. We implemented extensive quality control at several steps in our analysis in order to exclude lower quality data, however we still acknowledge the potential impact of remaining subtle motion not directly visible by the human eye, termed as "micro-motion" in Alexander-Bloch et al., 2016.
2. Since such subtle motion is not visible by eye, there is no way to assess whether this potential bias would affect pre- and postnatal scans differently. For instance, one could argue that shorter acquisition time for T2w slices combined with the super-resolution technique used for fetal scans could mitigate the potential influence of subtle motion for fetal data. In the present study, there is no way to control for the potential effect of subtle motion.
3. To the best of our knowledge, the potential effect of subtle motion has never been considered nor acknowledged in any prior study involving fetal MRI. For instance, the recent large-scale study brain-chart by Bethlehem et al., 2022 did not mention this factor in their limitations. This does not mean that we should ignore this potential bias, but we argue that this limitation is likely to affect the entire literature of fetal brain MRI.
4. In Alexander-Bloch et al., 2016, the authors observe that "in terms of the estimated effect of a standard deviation change in age or a standard deviation change in micro-motion, the estimated effect of age on morphology was on average approximately three times that of the effect size of micro-motion". In our study, the increase in gyrification at birth is as large as 50% of the total increase with age during the defined fetal period (between 21 and 37 wPC). Therefore it is very unlikely that the effect of subtle motion could explain the sharp increase in gyrification between fetal and postnatal data.

In addition, we apologize for our lack of clarity but we did not intend to claim that global measures are not affected by subtle motion. We argue that since the impact is heterogeneous in space, the impact on global measures is lower due to the averaging of local effects across brain regions in global measures. This is consistent with the reported statistics from Table 2 of Alexander-Bloch et al., 2016 with lower values for global measures than at the lobar scale.

In order to take this concern into account in the revised manuscript, we modified the following sentence in the limitations (**see page 21, line 439**).

"Furthermore, it is important to note that remaining subtle motion not directly visible by the human eye, termed as "micro-motion" in Alexander-Bloch et al., 2016, might affect our results, as well as previous literature on fetal brain MRI."

- my original point 5: the code is not available in the github provided. the files names do not have links.

We apologize for this unexpected issue. This is probably due to the "." at the end of the sentence that was included in the link. We have double checked and confirmed that the github scripts are accessible.